# Exploiting the Surrogate Gap in Online Multiclass Classification

**Dirk van der Hoeven**
Mathematical Institute
Leiden University
dirk@dirkvanderhoeven.com

## Abstract

We present GAPTRON, a randomized first-order algorithm for online multiclass classification. In the full information setting we provide expected mistake bounds for GAPTRON with respect to the logistic loss, hinge loss, and the smooth hinge loss with $O(K)$ regret, where the expectation is with respect to the learner's randomness and $K$ is the number of classes. In the bandit classification setting we show that GAPTRON is the first linear time algorithm with $O(K\sqrt{T})$ expected regret. Additionally, the expected mistake bound of GAPTRON does not depend on the dimension of the feature vector, contrary to previous algorithms with $O(K\sqrt{T})$ regret in the bandit classification setting. We present a new proof technique that exploits the gap between the zero-one loss and surrogate losses rather than exploiting properties such as exp-concavity or mixability, which are traditionally used to prove logarithmic or constant regret bounds.

## 1  Introduction

In online multiclass classification a learner has to repeatedly predict the label that corresponds to a feature vector. Algorithms in this setting have a wide range of applications ranging from predicting the outcomes of sport matches to recommender systems. In some applications such as sport forecasting the learner obtains the true label regardless of what outcome the learner predicts, but in other applications such as recommender systems the learner only learns whether or not the label he predicted was the true label. The setting in which the learner receives the true label is called the full information multiclass classification setting and the setting in which the learner only receives information about the predicted label is called the bandit multiclass classification setting.

In this paper we consider both the full information and bandit multiclass classification settings. In both settings the environment chooses the true outcome $y_t \in \{1, \ldots, K\}$ and feature vector $\boldsymbol{x}_t \in \mathbb{R}^d$. The environment then reveals the feature vector to the learner, after which the learner issues a (randomized) prediction $y_t' \in \{1, \ldots, K\}$. The goal of both settings is to minimize the number of expected mistakes the learner makes with respect to the best offline linear predictor $\boldsymbol{U} \in \mathbb{R}^{K \times d}$, which essentially contains a set of parameters per class. Standard practice in both settings is to upper bound the non-convex zero-one loss with a convex surrogate loss $\ell_t$ (see for example Bartlett et al. (2006)). This leads to guarantees of the form

$$\mathbb{E}\left[\sum_{t=1}^{T} \mathbb{1}[y_t' \neq y_t]\right] = \mathbb{E}\left[\sum_{t=1}^{T} \ell_t(\boldsymbol{U})\right] + \mathcal{R}_T,$$

where $\mathbb{1}$ is the indicator function, $y_t$ is the true label, the expectation is taken with respect to the learner's randomness, and $\mathcal{R}_T$ is the regret after $T$ rounds.

We introduce GAPTRON, which is a randomized first-order algorithm that exploits the gap between the zero-one loss and the surrogate loss. In the full information multiclass classification setting GAPTRON

Table 1: Main results and comparisons with previous work (see Section 2 for notation). The references are for the regret bounds, not necessarily for the first analysis of the algorithm. For this table we assume that $\|\boldsymbol{x}_t\| \le 1 \; \forall t$ and denote by $L_T = \sum_{t=1}^{T} \ell_t(\boldsymbol{U})$ the sum of the surrogate losses of the comparator.

| Algorithm | Loss | Regret full information setting | Regret bandit setting | Time (per round) |
|---|---|---|---|---|
| PERCEPTRON(Fink et al., 2006; Kakade et al., 2008) | hinge | $O(\|\boldsymbol{U}\|^2 + \|\boldsymbol{U}\|\sqrt{L_T})$ | $O((DK)^{1/3}T^{2/3})$ | $O(dK)$ |
| Second-Order PERCEPTRON (Orabona et al., 2012; Beygelzimer et al., 2017) | hinge[2] | $O(\frac{\kappa}{2-\kappa}\|\boldsymbol{U}\|^2 + \frac{dK}{\kappa(2-\kappa)}\ln(L_T))$ | $O(\|\boldsymbol{U}\|^2 + \frac{K}{\kappa}\sqrt{dT\ln(T)})$ | $O((dK)^2)$ |
| ONS (Hazan et al., 2014; Hazan and Kale, 2011) | logistic | $O(\exp(D)dK\ln(T))$ | $O(dK^3DT^{2/3})$ | $O((dK)^2)$ |
| Vovk's Aggregating Algorithm (Foster et al., 2018) | logistic | $O(dK\ln(DT))$ | $O(K\sqrt{dT\ln(DT)})$ | $O(\max\{dK,T\}^{12})$ |
| GAPTRON (This work) | logistic, hinge, smooth hinge | $O(K\|\boldsymbol{U}\|^2)$ | $O(KD\sqrt{T})$ | $O(dK)$ |

has $O(K)$ regret with respect to various surrogate losses. In the bandit multiclass classification setting we show that GAPTRON has $O(K\sqrt{T})$ regret with respect to the same surrogate losses as in the full information setting. Importantly, our regret bounds do not depend on the dimension of the feature vector in either the full or bandit information setting, contrary to previous results with similar regret bounds. Furthermore, in the bandit multiclass classification setting GAPTRON is the first $O(dK)$ running time algorithm with $O(K\sqrt{T})$ regret.

To achieve these results we develop a new proof technique. Standard approaches that lead to small regret bounds exploit properties of the surrogate loss function such as strong convexity, exp-concavity (Hazan et al., 2007), or mixability (Vovk, 2001). Instead, inspired by the recent success of Neu and Zhivotovskiy (2020) in online classification with abstention[1], we exploit the *gap* between the zero-one loss, which is used to measure the performance of the learner, and the surrogate loss, which is used to measure the performance of the comparator $\boldsymbol{U}$, hence the name GAPTRON.

For an overview of our results and a comparison to previous work see Table 1. Here we briefly discuss the most relevant literature to place our results into perspective. A more detailed comparison can be found in the relevant sections. The full information multiclass classification setting is well understood and has been studied by many authors. Perhaps the most well known algorithm in this setting is the PERCEPTRON (Rosenblatt, 1958) and its multiclass versions (Crammer and Singer, 2003; Fink et al., 2006). The PERCEPTRON is a deterministic first-order algorithm which has $O(\sqrt{T})$ regret with respect to the hinge loss in the worst-case. Variants of the PERCEPTRON such as AROW (Crammer et al., 2009) and the second-order PERCEPTRON (Cesa-Bianchi et al., 2005) are second-order methods which result in a possibly smaller regret at the cost of longer running time. Online logistic regression (Berkson, 1944) is an alternative to the PERCEPTRON which has been thoroughly studied. For an overview of results for online logistic regression we refer the reader to Shamir (2020). Here we mention a recent result by Foster et al. (2018), who use Exponential Weights (Vovk, 1990; Littlestone and Warmuth, 1994) to optimize the logistic loss and obtain a regret bound of $O(dK\ln(DT+1))$, where $D$ is an upper bound on the Frobenius norm of $\boldsymbol{U}$, with a polynomial time algorithm.

The bandit multiclass classification setting was first studied by Kakade et al. (2008) and is a special case of the contextual bandit setting (Langford and Zhang, 2008). Kakade et al. (2008) present a first-order algorithm called BANDITRON with a $O((DK)^{1/3}T^{2/3})$ regret bound with respect to the hinge loss. The impractical EXP4 algorithm (Auer et al., 2002) has a $O(\sqrt{TdK\ln(T+1)})$ regret bound and Abernethy and Rakhlin (2009) posed the problem of obtaining a practical algorithm which attains an $O(K\sqrt{T})$ regret bound. Several authors have proposed polynomial running time algorithms that have a regret bound of $O(K\sqrt{dT\log(T+1)})$ such as NEWTRON (Hazan and Kale, 2011), CONFIDIT (Crammer et al., 2009), SOBA (Beygelzimer et al., 2017), and OBAMA (Foster et al., 2018).

**Algorithm 1** GAPTRON
---
**Input:** Learning rate $\eta > 0$, exploration rate $\gamma \in [0, 1]$, and gap map $a : \mathbb{R}^{K \times d} \times \mathbb{R}^d \to [0, 1]$
1: **Initialize** $\boldsymbol{W}_1 = \boldsymbol{0}$
2: **for** $t = 1 \dots T$ **do**
3:     Obtain $\boldsymbol{x}_t$
4:     Let $y_t^\star = \arg \max_k \langle \boldsymbol{W}_t^k, \boldsymbol{x}_t \rangle$
5:     Set $\boldsymbol{p}_t' = (1 - \max\{a(\boldsymbol{W}_t, \boldsymbol{x}_t), \gamma\}) \boldsymbol{e}_{y_t^\star} + \max\{a(\boldsymbol{W}_t, \boldsymbol{x}_t), \gamma\} \frac{1}{K} \boldsymbol{1}$
6:     Predict with label $y_t' \sim \boldsymbol{p}_t'$
7:     Obtain $\mathbb{1}[y_t' \neq y_t]$ and set $\boldsymbol{g}_t = \nabla \ell_t(\boldsymbol{W}_t)$
8:     Update $\boldsymbol{W}_{t+1} = \arg \min_{\boldsymbol{W} \in \mathcal{W}} \eta \langle \boldsymbol{g}_t, \boldsymbol{W} \rangle + \frac{1}{2} \|\boldsymbol{W} - \boldsymbol{W}_t\|^2$
9: **end for**
---

## 2 Preliminaries

**Notation.**    Let $\boldsymbol{1}$ and $\boldsymbol{0}$ denote vectors with only ones and zeros respectively and let $\boldsymbol{e}_k$ denote the basis vector in direction $k$. The inner product between vectors $\boldsymbol{g} \in \mathbb{R}^d$ and $\boldsymbol{w} \in \mathbb{R}^d$ is denoted by $\langle \boldsymbol{w}, \boldsymbol{g} \rangle$. The rows of matrix $\boldsymbol{W} \in \mathbb{R}^{K \times d}$ are denoted by $\boldsymbol{W}^1, \dots, \boldsymbol{W}^K$. We will interchangeably use $\boldsymbol{W}$ to denote a matrix and a column vector in $\mathbb{R}^{Kd}$ to avoid unnecessary notation. The vector form of matrix $\boldsymbol{W}$ is $(\boldsymbol{W}^1, \dots, \boldsymbol{W}^K)^\top$. The Frobenius norm of matrix $\boldsymbol{W}$ is denoted by $\|\boldsymbol{W}\| = \sqrt{\sum_{k=1}^K \sum_{i=1}^d W_{k,i}^2}$. Likewise the $l_2$ norm of vector $\boldsymbol{x}$ is denoted by $\|\boldsymbol{x}\| = \sqrt{\sum_{i=1}^d x_i^2}$. We denote the Kronecker product between matrices $\boldsymbol{W}$ and $\boldsymbol{U}$ by $\boldsymbol{W} \otimes \boldsymbol{U}$. For a given round $t$ we use $\mathbb{E}_t[\cdot]$ to denote the conditional expectation given the predictions $y_1', y_2', \dots, y_{t-1}'$.

### 2.1 Multiclass Classification

The multiclass classification setting proceeds in rounds $t = 1, \dots, T$. In each round $t$ the environment first picks an outcome $y_t \in \{1, \dots K\}$ and feature vector $\boldsymbol{x}_t$ such that $\|\boldsymbol{x}_t\| \leq X$ for all $t$. Before the learner makes his prediction $y_t'$ the environment reveals the feature vector $\boldsymbol{x}_t$ which the learner may use to form $y_t'$. In the full information multiclass classification setting, after the learner has issued $y_t'$, the environment reveals the outcome $y_t$ to the learner. In the bandit multiclass classification setting (Kakade et al., 2008) the environment only reveals whether the prediction of the learner was correct or not, i.e. $\mathbb{1}[y_t' \neq y_t]$. We only consider the adversarial setting, which means that we make no assumptions on how $y_t$ or $\boldsymbol{x}_t$ is generated. In both settings we allow the learner to use randomized predictions. The goal of the multiclass classification setting is to control the number of expected mistakes the learner makes in $T$ rounds: $M_T = \mathbb{E}\left[\sum_{t=1}^T \mathbb{1}[y_t' \neq y_t]\right]$, where the expectation is taken with respect to the learner's randomness.

Since the zero-one loss is non-convex a standard approach is to use a surrogate loss $\ell_t$ as a function of a weight matrix $\boldsymbol{W}_t \in \mathcal{W}$, where $\mathcal{W} = \{\boldsymbol{W} : \|\boldsymbol{W}\| \leq D\}$. The surrogate loss function is a convex upper bound on the zero-one loss, which is then optimized using an Online Convex Optimization algorithm such as Online Gradient Descent (OGD) (Zinkevich, 2003), Online Newton Step (ONS) (Hazan et al., 2007), or Exponential Weights (EW) (Vovk, 1990; Littlestone and Warmuth, 1994). In this paper we treat three surrogate loss functions: logistic loss, the hinge loss, and the smooth hinge loss, all of which result in different guarantees on the number of expected mistakes a learner makes.

## 3 GAPTRON

In this section we discuss GAPTRON (Algorithm 1). The prediction $y_t'$ is sampled from

$$\boldsymbol{p}_t' = (1 - \max\{a(\boldsymbol{W}_t, \boldsymbol{x}_t), \gamma\}) \boldsymbol{e}_{y_t^\star} + \max\{a(\boldsymbol{W}_t, \boldsymbol{x}_t), \gamma\} \frac{1}{K} \boldsymbol{1},$$

where $\gamma \in [0, 1]$ and $a : \mathbb{R}^{K \times d} \times \mathbb{R}^d \to [0, 1]$ to be specified later. In the full information setting $\gamma$ is set to 0 but in the bandit setting $\gamma$ is used to guarantee that each label is sampled with at least probability $\frac{\gamma}{K}$, which is a common strategy in bandit algorithms (see for example Auer et al. (2002)). The fact that each label is sampled with at least probability $\frac{\gamma}{K}$ is important because in the bandit setting we use importance weighting to form estimated surrogate losses $\ell_t$ and their gradients

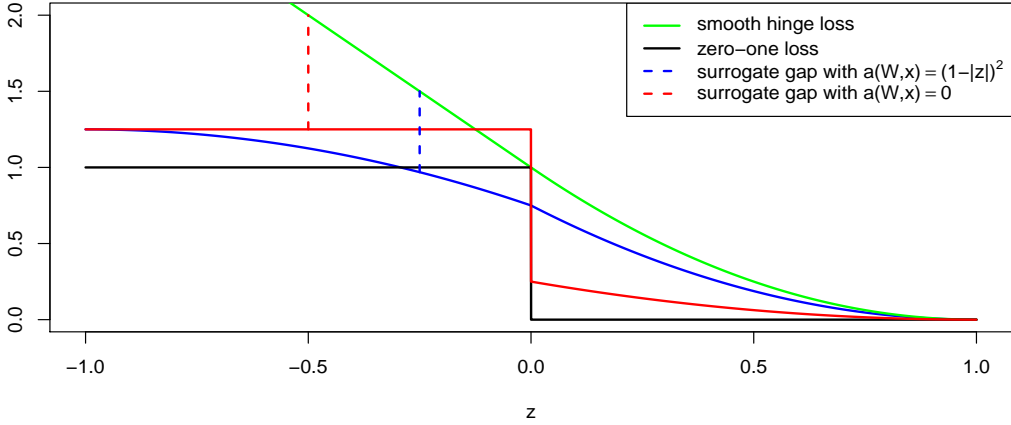

Figure 1: The surrogate gap for the smooth hinge loss as a function of margin $z$ with $\eta = \frac{1}{8}, \gamma = 0$, and $\|\boldsymbol{x}\| = 1$. The solid red line is given by $\mathbb{1}[z \leq 0] + \frac{\eta}{2}\|\boldsymbol{g}\|^2$, where $\|\boldsymbol{g}\|^2 = 4(1-z)^2$ if $z > 0$ and $\|\boldsymbol{g}\|^2 = 4$ otherwise. The solid blue line is given by $(1 - (1-|z|)^2)\mathbb{1}[z \leq 0] + \frac{1}{2}(1-|z|)^2 + \frac{\eta}{2}\|\boldsymbol{g}\|^2$. The surrogate gap is positive whenever the red or blue line is above the green line.

$\boldsymbol{g}_t = \nabla\ell_t(\boldsymbol{W}_t)$ and we need to control the variance of these estimates. The main difference between GAPTRON and standard algorithms for multiclass classification is the $a$ function, which governs the mixture that forms $\boldsymbol{p}'_t$. In fact, if we set $a(\boldsymbol{W}, \boldsymbol{x}) = 0$, $\gamma = 0$, and choose $\ell_t$ to be the hinge loss we recover an algorithm that closely resembles the PERCEPTRON (Rosenblatt, 1958), which can be interpreted as OGD on the hinge loss[3]. GAPTRON also uses OGD, which is used to update weight matrix $\boldsymbol{W}_t$, which in turn is used to form distribution $\boldsymbol{p}'_t$. For convenience we will define $a_t = a(\boldsymbol{W}_t, \boldsymbol{x}_t)$.

The role of $a$, which we will refer to as the gap map, is to exploit the gap between the surrogate loss and the zero-one loss. Before we explain how we exploit said gap we first present the expected mistake bound of GAPTRON in Lemma 1. The proof of Lemma 1 follows from applying the regret bound of OGD and working out the expected number of mistakes. The formal proof can be found in Appendix B.

**Lemma 1.** *For any $\boldsymbol{U} \in \mathcal{W}$ Algorithm 1 satisfies*

$$\mathbb{E}\left[\sum_{t=1}^{T}\mathbb{1}[y'_t \neq y_t]\right] \leq \mathbb{E}\left[\sum_{t=1}^{T}\ell_t(\boldsymbol{U})\right] + \frac{\|\boldsymbol{U}\|^2}{2\eta} + \gamma\frac{K-1}{K}T$$
$$+ \sum_{t=1}^{T}\underbrace{\mathbb{E}\left[(1-a_t)\mathbb{1}[y_t^\star \neq y_t] + a_t\frac{K-1}{K} - \ell_t(\boldsymbol{W}_t) + \frac{\eta}{2}\|\boldsymbol{g}_t\|^2\right]}_{\text{surrogate gap}}.$$

As we mentioned before, standard classifiers such as the PERCEPTRON simply set $a(\boldsymbol{W}, \boldsymbol{x}) = 0$ and upper bound $\mathbb{1}[y_t^\star \neq y_t] - \ell_t(\boldsymbol{W}_t)$ by 0. In the full information setting we can set $\gamma = 0$ and $\eta = \sqrt{\frac{\|\boldsymbol{U}\|^2}{\sum_{t=1}^{T}\|\boldsymbol{g}_t\|^2}}$ to obtain[4] $M_T \leq \sum_{t=1}^{T}\ell_t(\boldsymbol{U}) + \|\boldsymbol{U}\|\sqrt{\sum_{t=1}^{T}\|\boldsymbol{g}_t\|^2}$. However, the gap between the surrogate loss and the zero-one loss can be large. In fact, even with $a(\boldsymbol{W}, \boldsymbol{x}) = 0$, the gap between the zero-one loss and the surrogate loss is large enough to bound $\mathbb{1}[y_t^\star \neq y_t] - \ell_t(\boldsymbol{W}_t) + \frac{\eta}{2}\|\boldsymbol{g}_t\|^2$ by 0 for some loss functions and values of $\boldsymbol{W}_t$ and $\boldsymbol{x}_t$.

In Figure 1 we can see a depiction of the surrogate gap for the smooth hinge loss for $K = 2$ (Rennie and Srebro, 2005) in the full information setting (see Section 4.3 for the definition of the smooth

multiclass hinge loss). In the case where $K = 2$, $\boldsymbol{W}$ is a vector rather than a matrix and outcomes $y_t$ are coded as $\{-1, 1\}$. We see that with $a(W, \boldsymbol{x}) = 0$, only when margin $z = y\langle \boldsymbol{W}, \boldsymbol{x} \rangle \in [-0.125, 0]$ the surrogate gap is not upper bounded by 0. Decreasing $\eta$ would increase the range for which the surrogate gap is bounded by zero, but only for $\eta = 0$ the surrogate gap is bounded by 0 everywhere. However, with $a(W, \boldsymbol{x}) = (1 - |z|)^2$ the surrogate gap is upper bounded by 0 for all $z$, which leads to $O(K)$ regret. The remainder of the paper is concerned with deriving different $a$ for different loss functions for which the surrogate gap is bounded by 0. For an overview of the different settings of GAPTRON see Table 2 in Appendix A. In the following section we start in the full information setting.

## 4 Full Information Multiclass Classification

In this section we derive gap maps that allow us to upper bound the surrogate gap by 0 for the logistic loss, hinge loss, and smooth hinge loss in the full information setting. Note that the full information algorithms we compare with are deterministic whereas GAPTRON is randomized. Throughout this section we will set $\gamma = 0$. We start with the result for logistic loss.

### 4.1 Logistic Loss

The logistic loss is defined as

$$\ell_t(\boldsymbol{W}) = -\log_2(\sigma(\boldsymbol{W}, \boldsymbol{x}_t, y_t)), \tag{1}$$

where $\sigma(\boldsymbol{W}, \boldsymbol{x}, k) = \frac{\exp(\langle \boldsymbol{W}^k, \boldsymbol{x} \rangle)}{\sum_{\tilde{k}=1}^K \exp(\langle \boldsymbol{W}^{\tilde{k}}, \boldsymbol{x} \rangle)}$ is the softmax function. For the logistic loss we will use the following gap map:

$$a(\boldsymbol{W}_t, \boldsymbol{x}_t) = 1 - \mathbb{1}[p_t^\star \geq 0.5]p_t^\star,$$

where $p_t^\star = \max_k \sigma(\boldsymbol{W}_t, \boldsymbol{x}_t, k)$. This means that GAPTRON samples a label uniformly at random as long as $p_t^\star \leq 0.5$. While this may appear counter-intuitive at first sight note that when $p_t^\star < 0.5$ the zero-one loss is upper bounded by the logistic loss regardless of what we play since $-\log_2(p) \geq 1$ for $p \in [0, 0.5]$, which we use to show that the surrogate gap is bounded by 0 whenever $p_t^\star < 0.5$. The mistake bound of GAPTRON can be found in Theorem 1. To prove Theorem 1 we show that the surrogate gap is bounded by 0 and then use Lemma 1. The formal proof can be found in Appendix C.1.

**Theorem 1.** *Let* $a(\boldsymbol{W}_t, \boldsymbol{x}_t) = 1 - \mathbb{1}[p_t^\star \geq 0.5]p_t^\star$, $\eta = \frac{\ln(2)}{2KX^2}$, $\gamma = 0$, *and let* $\ell_t$ *be the logistic loss defined in* (1). *Then for any* $\boldsymbol{U} \in \mathcal{W}$ *Algorithm 1 satisfies*

$$\mathbb{E}\left[\sum_{t=1}^T \mathbb{1}[y_t' \neq y_t]\right] \leq \sum_{t=1}^T \ell_t(\boldsymbol{U}) + \frac{KX^2\|\boldsymbol{U}\|^2}{\ln(2)}.$$

Let us compare the mistake bound of GAPTRON with other results for logistic loss. Foster et al. (2018) circumvent a lower bound for online logistic regression by Hazan et al. (2014) by using an improper learning algorithm and achieve $O(dK \ln(DT + 1))$ regret. Unfortunately this algorithm is impractical since the running time can be of $O(D^6 \max\{dK, T\}^{12}T)$. In the case where $K = 2$ Jézéquel et al. (2020) provide a faster improper learning algorithm called AIOLI based on the Vovk-Azoury-Warmuth forecaster (Vovk, 2001; Azoury and Warmuth, 2001) that has running time $O(d^2T)$ and a regret of $O(dD \ln(T))$. Unfortunately it is not known if AIOLI can be extended to $K > 2$. An alternative algorithm is ONS, which has running time $O((dK)^2T)$ but a regret bound of $O(\exp(D)dK \ln(T + 1))$. With standard OGD we could degrade the dependence on $T$ to improve the dependence on $D$ to find a regret of $O(D\sqrt{T})$ with an algorithm that has running time $O(dKT)$. Depending on $\|\boldsymbol{U}\|^2$ the regret of GAPTRON can be significantly smaller than the regret of the aforementioned algorithms as the regret of GAPTRON is independent of $T$ and $d$. Furthermore, since GAPTRON uses OGD to update $\boldsymbol{W}_t$ the running time is $O(dKT)$, significantly improving upon the running time of previous algorithms with comparable mistake bounds.

## 4.2 Multiclass Hinge Loss

We use a variant of the multiclass hinge loss of Crammer and Singer (2001), which is defined as:

$$\ell_t(\boldsymbol{W}) = \begin{cases} \max\{1 - m_t(\boldsymbol{W}, y_t), 0\} & \text{if } m_t^\star \leq \beta \\ \max\{1 - m_t(\boldsymbol{W}, y_t), 0\} & \text{if } y_t^\star \neq y_t \text{ and } m_t^\star > \beta \\ 0 & \text{if } y_t^\star = y_t \text{ and } m_t^\star > \beta, \end{cases} \quad (2)$$

where $m_t(\boldsymbol{W}_t, y) = \langle \boldsymbol{W}_t^y, \boldsymbol{x}_t \rangle - \max_{k \neq y}\langle \boldsymbol{W}_t^k, \boldsymbol{x}_t \rangle$ and $m_t^\star = \max_k m_t(\boldsymbol{W}_t, k)$. Note that we set $\ell_t(\boldsymbol{W}) = 0$ when $y_t^\star = y_t$ and $m_t^\star > \beta$. In common implementations of the PERCEPTRON $\ell_t(\boldsymbol{W}) = 0$ whenever $y_t^\star = y_t$ (see for example Kakade et al. (2008)). However, for the surrogate gap to be bounded by zero we need $\ell_t$ to be positive whenever $a_t > 0$ otherwise there is nothing to cancel out the $a_t \frac{K-1}{K}$ term. The gap map for the hinge loss is $a(W_t, \boldsymbol{x}_t) = 1 - \max\{\mathbb{1}[m_t^\star > \beta], m_t^\star\}$. This means that whenever $m_t^\star > \beta$ the predictions of GAPTRON are identical to the predictions of the PERCEPTRON. The mistake bound of GAPTRON for the hinge loss can be found in Theorem 2 (its proof is deferred to Appendix C.2).

**Theorem 2.** *Set* $a(\boldsymbol{W}_t, \boldsymbol{x}_t) = 1 - \max\{\mathbb{1}[m_t^\star > \beta], m_t^\star\}$, $\eta = \frac{1-\beta}{KX^2}$, $\gamma = 0$, *and let* $\ell_t$ *be the multiclass hinge loss defined in* (2) *with* $\beta = \frac{1}{K}$. *Then for any* $\boldsymbol{U} \in \mathcal{W}$ *Algorithm 1 satisfies*

$$\mathbb{E}\left[\sum_{t=1}^T \mathbb{1}[y_t' \neq y_t]\right] \leq \sum_{t=1}^T \ell_t(\boldsymbol{U}) + \frac{K^2 X^2 \|\boldsymbol{U}\|^2}{2(K-1)}.$$

Let us compare the mistake bound of GAPTRON with the mistake bound of the PERCEPTRON. The PERCEPTRON guarantees $M_T \leq \sum_{t=1}^T \ell_t(\boldsymbol{U}) + X^2\|\boldsymbol{U}\|^2 + 2X\|\boldsymbol{U}\|\sqrt{2\sum_{t=1}^T \ell_t(\boldsymbol{U})}$ (see Beygelzimer et al. (2017) for a proof). The factor $K$ in the regret of GAPTRON is due to the cost of exploring uniformly at random. For small $K$ the mistake bound of GAPTRON can be significantly smaller in the adversarial case, but for large $K$ the cost of sampling uniformly at random can be too high and the mistake bound of GAPTRON can be larger than that of the PERCEPTRON. In the separable case the PERCEPTRON has a strictly better guarantee for any $K$ since then only the $X^2\|\boldsymbol{U}\|^2$ term remains.

Orabona et al. (2012) show that for all loss functions of the form $\ell_t(\boldsymbol{W}) = \max\{1 - \frac{2}{2-\kappa}m_t(W, y_t) + \frac{\kappa}{2-\kappa}m_t(W, y_t)^2, 0\}$ the second-order PERCEPTRON guarantees $M_T \leq \sum_{t=1}^T \ell_t(\boldsymbol{U}) + O(\frac{\kappa}{2-\kappa}X^2\|\boldsymbol{U}\|^2 + \frac{dK}{\kappa(2-\kappa)}\ln(\sum_{t=1}^T \ell_t(\boldsymbol{U}) + 1))$. Thus, for small $K$ GAPTRON always has a smaller regret term but for larger $K$ the guarantee of GAPTRON can be worse, although this also depends on the performance and norm of the comparator $\boldsymbol{U}$.

## 4.3 Smooth Multiclass Hinge Loss

The smooth multiclass hinge loss (Rennie and Srebro, 2005) is defined as

$$\ell_t(\boldsymbol{W}) = \begin{cases} \max\{1 - 2m_t(\boldsymbol{W}, y_t), 0\} & \text{if } m_t(\boldsymbol{W}, y_t) \leq 0 \\ \max\{(1 - m_t(\boldsymbol{W}, y_t))^2, 0\} & \text{if } m_t(\boldsymbol{W}, y_t) > 0, \end{cases} \quad (3)$$

where $m_t(\boldsymbol{W}_t, y) = \langle \boldsymbol{W}_t^y, \boldsymbol{x}_t \rangle - \max_{k \neq y}\langle \boldsymbol{W}_t^k, \boldsymbol{x}_t \rangle$ as in Section 4.3. This loss function is not exp-concave nor is it strongly-convex. This means that with standard methods from Online Convex Optimization we cannot hope to achieve a better regret bound than $O(D\sqrt{T})$ in the worst-case. Theorem 3 shows that with gap map $a(\boldsymbol{W}_t, x_t) = (1 - \min\{1, m_t^\star\})^2$, where $m_t^\star = \max_k m_t(\boldsymbol{W}_t, k)$, GAPTRON has an $O(K)$ regret bound. The proof of Theorem 3 follows from bounding the surrogate gap by zero and can be found in Appendix C.3.

**Theorem 3.** *Set* $a(\boldsymbol{W}_t, \boldsymbol{x}_t) = (1 - \min\{1, m_t^\star\})^2$, $\eta = \frac{1}{4KX^2}$, $\gamma = 0$, *and let* $\ell_t$ *be the smooth multiclass hinge loss defined in* (3). *Then for any* $\boldsymbol{U} \in \mathcal{W}$ *Algorithm 1 satisfies*

$$\mathbb{E}\left[\sum_{t=1}^T \mathbb{1}[y_t' \neq y_t]\right] \leq \sum_{t=1}^T \ell_t(\boldsymbol{U}) + 2KX^2\|\boldsymbol{U}\|^2.$$

# 5 Bandit Multiclass Classification

In this section we will analyse GAPTRON in the bandit multiclass classification setting. While in the full information setting the fact that GAPTRON is a randomized algorithm can be seen as a drawback, in the adversarial bandit setting it is actually a requirement (see for example chapter 11 by Lattimore and Szepesvári (2020)). We will use the same gap maps as in the full information setting. The only difference is how we feed the surrogate loss to GAPTRON. We will use the same loss functions as in the full information setting but now multiplied by $\frac{\mathbb{1}[y_t'=y_t]}{p_t'(y_t')}$, which is simply importance weighting. This also means that, compared to the full information setting, the gradients that OGD uses to update weight matrix $\boldsymbol{W}_t$ are multiplied by $\frac{\mathbb{1}[y_t'=y_t]}{p_t'(y_t')}$. To control the surrogate gap we set $\gamma > 0$, which allows us to bound the variance of the norm of the gradients. The proofs in this section follow the same structure as in the full information setting, with the notable change that we suffer increased regret due to the $\gamma \frac{K-1}{K} T$ bias term and the increased $\mathbb{E}[\|\boldsymbol{g}_t\|^2] = O(\frac{K}{\gamma})$ term.

The results in this section provide three new answers to the open problem by Abernethy and Rakhlin (2009), who posed the problem of obtaining an efficient algorithm with $O(K\sqrt{T})$ regret. Several solutions with various loss function have been proposed. Beygelzimer et al. (2017) solved the open problem using an algorithm called SOBA. SOBA is a second-order algorithm which is analysed using a family of surrogate loss functions introduced by Orabona et al. (2012) ranging from the standard multiclass hinge loss to the squared multiclass hinge loss. The loss functions are parameterized by $\kappa$, where $\kappa = 0$ corresponds to the multiclass hinge loss and $\kappa = 1$ corresponds to the squared hinge loss. Simultaneously for all surrogate loss functions in the family of loss functions SOBA suffers a regret of $O(\|\boldsymbol{U}\|^2 X^2 + \frac{K}{\kappa}\sqrt{dT \ln(T+1)})$ and has a running time of order $O((dK)^2 T)$. Hazan and Kale (2011) consider the logistic loss and obtain regret of $O(dK^3 \min\{\exp(DX)\ln(T+1), DXT^{\frac{2}{3}}\})$. Hazan and Kale (2011) also obtain $DX\sqrt{T}$ regret for a variant of the logistic loss function we consider in this paper. Both results of Hazan and Kale (2011) are obtained by running ONS on (a variant of) the logistic loss, which has running time $O((dK)^2 T)$. Crammer et al. (2009) assume that a particular probabilistic model generates $y_t$ and obtain $O(DK\sqrt{dT}\log(T))$ expected regret bound with high probability, but for a sharper notion of regret. Foster et al. (2018) introduce OBAMA, which improves the results of Hazan and Kale (2011) and suffers $O(\min\{dK^2 \ln(TDX+1), K\sqrt{dT \ln(TDX+1)}\})$ regret for the logistic loss. Unfortunately, OBAMA has running time $O(D^6 \max\{dK, T\}^{12} T)$.

GAPTRON is the first $O(dKT)$ running time algorithm which has $O(DK\sqrt{T})$ regret in bandit multiclass classification with respect to the logistic, hinge, or smooth hinge loss. GAPTRON also improves the regret bounds of previous algorithms with $O(DK\sqrt{T})$ regret by a factor $O(\sqrt{d\log(T+1)})$. The remainder of this section provides the settings for GAPTRON to achieve these results, starting with the logistic loss.

## 5.1 Bandit Logistic Loss

The bandit version of the logistic loss is defined as:

$$\ell_t(\boldsymbol{W}) = -\frac{\mathbb{1}[y_t' = y_t]}{p_t'(y_t')} \log_2(\sigma(\boldsymbol{W}, \boldsymbol{x}_t, y_t)). \tag{4}$$

A similar definition of the bandit logistic loss is used by Hazan and Kale (2011); Foster et al. (2018). It is straightforward to verify that $\mathbb{E}_t[\ell_t(\boldsymbol{w})]$ is equivalent to its full information counterpart (1). This loss is a factor $\frac{1}{\ln(2)}$ larger than the loss used by Hazan and Kale (2011); Foster et al. (2018), who use the natural logarithm instead of the logarithm with base 2. To stay consistent with the full information setting we opt to use base 2 in the bandit setting. Using GAPTRON with the natural logarithm will give similar results.

The mistake bound of GAPTRON for this loss can be found in Theorem 4 (its proof can be found in Appendix D.1). Compared to OBAMA, which achieves a regret bound of $O(\min\{dK^2 \ln(TDX+1), K\sqrt{dT \ln(TDX+1)}\})$, GAPTRON has a larger dependency on $D$ and $X$. However, the mistake bound of GAPTRON does not depend on $d$, which can be a significant improvement over the regret bound of OBAMA. Theorem 4 answers the two questions by Hazan and

Kale (2011) affirmatively; GAPTRON is a linear time algorithm with exponentialy improved constants in the regret bound compared to NEWTRON.

**Theorem 4.** *Let* $a(\boldsymbol{W}_t, \boldsymbol{x}_t) = 1 - \mathbb{1}[p_t^\star \geq 0.5]p_t^\star$, $\eta = \frac{\ln(2)((1-\gamma)\exp(-2DX)\frac{1}{K}+\gamma)}{2K^2X^2}$, *and let* $\ell_t$ *be the bandit logistic loss* (4). *Then there exists a setting of* $\gamma$ *such that Algorithm 1 satisfies*

$$\mathbb{E}\left[\sum_{t=1}^T \mathbb{1}[y_t' \neq y_t]\right] \leq \mathbb{E}\left[\sum_{t=1}^T \ell_t(\boldsymbol{U})\right] + KXD \min\left\{\max\left\{\frac{2KXD}{\ln(2)}, 2\sqrt{\frac{T}{\ln(2)}}\right\}, \frac{KXD}{e^{-2DX}\ln(2)}\right\}.$$

## 5.2 Bandit Multiclass Hinge Loss

We use the following definition of the bandit multiclass hinge loss:

$$\ell_t(\boldsymbol{W}_t) = \begin{cases} \frac{\mathbb{1}[y_t'=y_t]}{p_t'(y_t')}\max\{1 - m_t(\boldsymbol{W}_t, y_t), 0\} & \text{if } m_t^\star \leq \beta \\ \frac{\mathbb{1}[y_t'=y_t]}{p_t'(y_t')}\max\{1 - m_t(\boldsymbol{W}_t, y_t), 0\} & \text{if } y_t^\star \neq y_t \text{ and } m_t^\star > \beta \\ 0 & \text{if } y_t' = y_t^\star = y_t \text{ and } m_t^\star > \beta. \end{cases} \tag{5}$$

It is straightforward to see that the conditional expectation of the bandit multiclass hinge loss is the full information multiclass hinge loss. Both the BANDITRON algorithm (Kakade et al., 2008) and SOBA (Beygelzimer et al., 2017) use a similar loss function.

As we mentioned before, Beygelzimer et al. (2017) present SOBA, which is a second-order algorithm with regret $O(\|\boldsymbol{U}\|^2 X^2 + \frac{K}{\kappa}\sqrt{dT\ln(T+1)})$. BANDITRON is a first-order algorithm based on the PERCEPTRON algorithm and suffers $O((KDX)^{1/3}T^{2/3})$ regret. For the more general setting of contextual bandits (Foster and Krishnamurthy, 2018) use continuous Exponential Weights with the hinge loss to also obtain an $O(KDX\sqrt{dT\ln(T+1)})$ regret bound with a polynomial time algorithm. The expected mistake bound of GAPTRON can be found in Theorem 5 and its proof can be found in Appendix D.2. Compared to the BANDITRON GAPTRON has larger regret in terms of $D$, $K$, and $X$, but smaller regret in terms of $T$. Compared to the regret of SOBA the regret of GAPTRON does not contain a factor $\sqrt{d\ln(T+1)}$.

**Theorem 5.** *Set* $a(\boldsymbol{W}_t, \boldsymbol{x}_t) = 1 - \max\{\mathbb{1}[m_t^\star > \beta], m_t^\star\}$, $\eta = \frac{\gamma(1-\beta)}{K^2X^2}$, $\gamma = \min\left\{1, \sqrt{\frac{K^3X^2D^2}{2(1-\beta)(K-1)T}}\right\}$, *and let* $\ell_t$ *be the bandit multiclass hinge loss defined in* (5) *with* $\beta = \frac{1}{K}$. *Then for any* $\boldsymbol{U} \in \mathcal{W}$ *Algorithm 1 satisfies*

$$\mathbb{E}\left[\sum_{t=1}^T \mathbb{1}[y_t' \neq y_t]\right] \leq \mathbb{E}\left[\sum_{t=1}^T \ell_t(\boldsymbol{U})\right] + \max\left\{\frac{K^3X^2D^2}{K-1}, 2KXD\sqrt{\frac{T}{2}}\right\}.$$

## 5.3 Bandit Smooth Multiclass Hinge Loss

In this section we use the following loss function:

$$\ell_t(\boldsymbol{W}) = \begin{cases} \frac{\mathbb{1}[y_t'=y_t]}{p_t'(y_t)}\max\{1 - 2m_t(\boldsymbol{W}, y_t), 0\} & \text{if } m_t(\boldsymbol{W}, y_t) \leq 0 \\ \frac{\mathbb{1}[y_t'=y_t]}{p_t'(y_t)}\max\{(1 - m_t(\boldsymbol{W}, y_t))^2, 0\} & \text{if } m_t(\boldsymbol{W}, y_t) > 0. \end{cases} \tag{6}$$

This loss function is the bandit version of the smooth multiclass hinge loss that we we used in Section 4.3 and its expectation is equivalent to its full information counterpart in equation (3). The regret of GAPTRON with this loss function can be found in Theorem 6. The proof of Theorem 6 can be found in Appendix D.3.

**Theorem 6.** *Set* $a(\boldsymbol{W}_t, \boldsymbol{x}_t) = (1 - \min\{1, m_t^\star\})^2$, $\eta = \frac{\gamma}{4K^2X^2}$, $\gamma = \min\left\{1, \sqrt{\frac{4K^2X^2D^2}{T}}\right\}$, *and let* $\ell_t$ *be the bandit smooth multiclass hinge loss defined in* (6). *Then for any* $\boldsymbol{U} \in \mathcal{W}$ *Algorithm 1 satisfies*

$$\mathbb{E}\left[\sum_{t=1}^T \mathbb{1}[y_t' \neq y_t]\right] \leq \mathbb{E}\left[\sum_{t=1}^T \ell_t(\boldsymbol{U})\right] + \max\left\{4K^2X^2D^2, 2KXD\sqrt{2T}\right\}.$$

## 6 Conclusion

In this paper we introduced GAPTRON, a randomized first-order algorithm for the full and bandit information multiclass classification settings. Using a new technique we showed that GAPTRON has an $O(K)$ regret bound in the full information setting and a regret bound of $O(K\sqrt{T})$ in the bandit setting. One of the main drawbacks of GAPTRON is that it is a randomized algorithm. Our bounds only hold in expectation and it would be interesting to show similar bounds also hold with high probability. Another interesting venue to explore is how to extend the ideas in this paper to the stochastic setting or the more general contextual bandit setting. In future work we would like to conduct experiments to compare GAPTRON with other algorithms, particularly in the bandit setting. Finally, as the results of Beygelzimer et al. (2019) show, in the separable bandit setting GAPTRON does not obtain the optimal regret bound. Deriving an algorithm that both has $O(K\sqrt{T})$ regret in the adversarial bandit setting and $O(K)$ regret in the separable bandit setting is also an interesting direction to pursue.

## 7 Broader Impact

Our contribution is primarily theoretical. Therefore, this work does not present any foreseeable societal consequence.

## Acknowledgments and Disclosure of Funding

The author would like to thank Tim van Erven and Sarah Sachs for their comments on an earlier version of this paper. The author would like to thank Francesco Orabona for pointing out two related references: Crammer and Gentile (2013) and Beygelzimer et al. (2019). The author was supported by the Netherlands Organization for Scientific Research (NWO grant TOP2EW.15.211).

## Footnotes

[1]In fact, in Appendix E we slightly generalize the results of Neu and Zhivotovskiy (2020).

[2]These results hold for a family of loss functions parametrized by $\kappa \in [0, 1]$, which includes the hinge loss.

[3]Other interpretations exist which lead to possibly better guarantees, see for example Beygelzimer et al. (2017).

[4]Although such tuning is impossible due to not knowing $\|\boldsymbol{U}\|$ or $\sum_{t=1}^{T}\|\boldsymbol{g}_t\|^2$ there exist algorithms that are able to achieve the same guarantee up to logarithmic factors, see for example Cutkosky and Orabona (2018).

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
