[Supplementary Material]

# A Table with Different Settings of GAPTRON

Table 2: Settings of Gaptron

| Surrogate Loss | Gap map $a$ | Learning rate $\eta$ | Exploration $\gamma$ | Regret |
|---|---|---|---|---|
| logistic (1) | $1 - \mathbb{1}[p_t^\star \geq 0.5]p_t^\star$ | $\frac{\ln(2)}{2KX^2}$ | $0$ | $\frac{KX^2\|U\|^2}{\ln(2)}$ |
| bandit logistic (4) | $1 - \mathbb{1}[p_t^\star \geq 0.5]p_t^\star$ | $\frac{\ln(2)\exp(-2DX)}{2K^2X^2}$ | $0$ | $\frac{\exp(2DX)K^2X^2D^2}{\ln(2)}$ |
| bandit logistic (4) | $1 - \mathbb{1}[p_t^\star \geq 0.5]p_t^\star$ | $\frac{\gamma\ln(2)}{2K^2X^2}$ | $\sqrt{\frac{2K^2X^2}{T}}$ | $2KXD\sqrt{\frac{T}{\ln(2)}}$ |
| hinge (2) | $1 - \max\{\mathbb{1}[m_t^\star > \beta]\}$ | $\frac{K-1}{K^2X^2}$ | $0$ | $\frac{K^2X^2\|U\|^2}{2(K-1)}$ |
| bandit hinge (5) | $1 - \max\{\mathbb{1}[m_t^\star > \beta]\}$ | $\frac{\gamma(K-1)}{K^3X^2}$ | $\sqrt{\frac{K^4X^2D^2}{2(K-1)^2T}}$ | $2KXD\sqrt{\frac{T}{2}}$ |
| smooth hinge (3) | $(1 - \min\{1, m_t^\star\})^2$ | $\frac{1}{2KX^2}$ | $0$ | $2KX^2\|U\|^2$ |
| bandit smooth hinge (6) | $(1 - \min\{1, m_t^\star\})^2$ | $\frac{\gamma}{2K^2X^2}$ | $\sqrt{\frac{4K^2X^2D^2}{T}}$ | $2DKX\sqrt{2T}$ |

# B Details of Section 3

*Proof of Lemma 1.* As we said before, the updates of $W_t$ are Online Gradient Descent (Zinkevich, 2003), which guarantees

$$\sum_{t=1}^{T} \ell_t(W_t) - \ell_t(U) \leq \frac{\|U\|^2}{2\eta} + \sum_{t=1}^{T} \frac{\eta}{2}\|g_t\|^2. \tag{7}$$

Now, by using (7) we find

$$\mathbb{E}\left[\sum_{t=1}^{T}\left(\mathbb{1}[y_t' \neq y_t] - \ell_t(U)\right)\right]$$

$$= \mathbb{E}\left[\sum_{t=1}^{T}\left(\mathbb{1}[y_t' \neq y_t]\right) - \ell_t(W_t)\right) + \sum_{t=1}^{T}\left(\ell_t(W_t) - \ell_t(U)\right)\right]$$

$$\leq \frac{\|U\|^2}{2\eta} + \mathbb{E}\left[\sum_{t=1}^{T}\left(\mathbb{E}_t[\mathbb{1}[y_t' \neq y_t]] - \ell_t(W_t) + \frac{\eta}{2}\|g_t\|^2\right)\right]$$

$$= \frac{\|U\|^2}{2\eta} + \mathbb{E}\left[\sum_{t=1}^{T}\left((1 - \max\{a_t, \gamma\})\mathbb{1}[y_t^\star \neq y_t] + \max\{a_t, \gamma\}\frac{K-1}{K} - \ell_t(W_t) + \frac{\eta}{2}\|g_t\|^2\right)\right]$$

$$\leq \frac{\|U\|^2}{2\eta} + \gamma\frac{K-1}{K}T + \mathbb{E}\left[\sum_{t=1}^{T}\left((1-a_t)\mathbb{1}[y_t^\star \neq y_t] + a_t\frac{K-1}{K} - \ell_t(W_t) + \frac{\eta}{2}\|g_t\|^2\right)\right], \tag{8}$$

where in the last inequality we used $(1 - \max\{a_t, \gamma\}) \leq (1 - a_t)$ and $\max\{a_t, \gamma\} \leq a_t + \gamma$. Adding $\mathbb{E}\left[\sum_{t=1}^{T} \ell_t(U)\right]$ to both sides of equation (8) completes the proof. $\qquad\square$

# C Details of Full Information Multiclass Classification (Section 4)

## C.1 Details of Section 4.1

*Proof of Theorem 1.* We will prove the Theorem by showing that the surrogate gap is bounded by 0 and then using Lemma 1. The gradient of the logistic loss evaluated at $W_t$ is given by:

$$\nabla\ell_t(W_t) = \frac{1}{\ln(2)}(\tilde{p}_t - e_{y_t}) \otimes x_t,$$

where $\tilde{p}_t = (\tilde{p}_t(1), \ldots, \tilde{p}_t(k))^\top$ and $\tilde{p}_t(k) = \sigma(W_t, x_t, k)$.

We continue by writing out the surrogate gap:

$$(1 - a_t)\mathbb{1}[y_t^\star \neq y_t] + a_t \frac{K-1}{K} - \ell_t(\boldsymbol{W}_t) + \frac{\eta}{2}\|\boldsymbol{g}_t\|^2$$

$$\leq (1 - a_t)\mathbb{1}[y_t^\star \neq y_t] + a_t \frac{K-1}{K} - \ell_t(\boldsymbol{W}_t) - \frac{\eta}{\ln(2)}\|\boldsymbol{x}_t\|^2 \log_2(\tilde{p}_t(y_t))$$

$$\leq (1 - a_t)\mathbb{1}[y_t^\star \neq y_t] + a_t \frac{K-1}{K} - \ell_t(\boldsymbol{W}_t) - \frac{\eta}{\ln(2)}X^2 \log_2(\tilde{p}_t(y_t)) \tag{9}$$

$$= \begin{cases} 0 + \frac{K-1}{K} + \log_2(\tilde{p}_t(y_t)) - \frac{\eta}{\ln(2)}X^2 \log_2(\tilde{p}_t(y_t)) & \text{if } p_t^\star < 0.5 \\ p_t^\star + (1 - p_t^\star)\frac{K-1}{K} + \log_2(\tilde{p}_t(y_t)) - \frac{\eta}{\ln(2)}X^2 \log_2(\tilde{p}_t(y_t)) & \text{if } y_t^\star \neq y_t \text{ and } p_t^\star \geq 0.5 \\ (1 - p_t^\star)\frac{K-1}{K} + \log_2(p_t^\star) - \frac{\eta}{\ln(2)}X^2 \log_2(p_t^\star) & \text{if } y_t^\star = y_t \text{ and } p_t^\star \geq 0.5, \end{cases}$$

where the first inequality is due to Lemma 2 below.

We now split the analysis into the cases in (9). We start with $p_t^\star < 0.5$. In this case we use $1 \leq -\log_2(x)$ for $x \in [0, \frac{1}{2}]$ and obtain

$$\frac{K-1}{K} + \log_2(\tilde{p}_t(y_t)) - \frac{\eta}{\ln(2)}X^2 \log_2(\tilde{p}_t(y_t))$$

$$\leq -\frac{K-1}{K}\log_2(\tilde{p}_t(y_t)) + \log_2(\tilde{p}_t(y_t)) - \frac{\eta}{\ln(2)}X^2 \log_2(\tilde{p}_t(y_t))$$

$$= \frac{1}{K}\log_2(\tilde{p}_t(y_t)) - \frac{\eta}{\ln(2)}X^2 \log_2(\tilde{p}_t(y_t)),$$

which is bounded by 0 since $\eta < \frac{\ln(2)}{KX^2}$.

The second case we consider is when $y_t^\star \neq y_t$ and $p_t^\star \geq 0.5$. In this case we use $x \leq -\frac{1}{2}\log_2(1-x)$ for $x \in [0.5, 1]$ and $1 - x \leq -\frac{1}{2}\log_2(1-x)$ for $x \in [0.5, 1]$ and obtain

$$p_t^\star + (1 - p_t^\star)\frac{K-1}{K} + \log_2(\tilde{p}_t(y_t)) - \frac{\eta}{\ln(2)}X^2 \log_2(\tilde{p}_t(y_t))$$

$$\leq -\frac{1}{2}\log_2(1 - p_t^\star) - \frac{K-1}{K}\frac{1}{2}\log_2(1 - p_t^\star) + \log_2(\tilde{p}_t(y_t)) - \frac{\eta}{\ln(2)}X^2 \log_2(\tilde{p}_t(y_t))$$

$$= -\frac{1}{2}\log_2\left(\sum_{k \neq y_t}^{K} \tilde{p}_t(k)\right) - \frac{K-1}{K}\frac{1}{2}\log_2\left(\sum_{k \neq y_t}^{K} \tilde{p}_t(k)\right) + \log_2(\tilde{p}_t(y_t)) - \frac{\eta}{\ln(2)}X^2 \log_2(\tilde{p}_t(y_t))$$

$$\leq -\frac{1}{2}\log_2(\tilde{p}_t(y_t)) - \frac{K-1}{K}\frac{1}{2}\log_2(\tilde{p}_t(y_t)) + \log_2(\tilde{p}_t(y_t)) - \frac{\eta}{\ln(2)}X^2 \log_2(\tilde{p}_t(y_t))$$

$$= \frac{1}{2K}\log_2(\tilde{p}_t(y_t)) - \frac{\eta}{\ln(2)}X^2 \log_2(\tilde{p}_t(y_t)),$$

which is 0 since $\eta = \frac{\ln(2)}{2KX^2}$.

The last case we need to consider is $y_t^\star = y_t$ and $p_t^\star \geq 0.5$. In this case we use $1 - x \leq -\log_2(x)$ and obtain

$$(1 - p_t^\star)\frac{K-1}{K} + \log_2(p_t^\star) - \frac{\eta}{\ln(2)}X^2 \log_2(p_t^\star)$$

$$\leq -\frac{K-1}{K}\log_2(p_t^\star) + \log_2(p_t^\star) - \frac{\eta}{\ln(2)}X^2 \log_2(p_t^\star),$$

which is bounded by 0 since $\eta = \frac{\ln(2)}{2KX^2}$.

We now apply Lemma 1, plug in $\gamma = 0$, and use the above to find:

$$\mathbb{E}\left[\sum_{t=1}^{T} \mathbb{1}[y_t' \neq y_t]\right] \leq \frac{\|\boldsymbol{U}\|^2}{2\eta} + \sum_{t=1}^{T} \ell_t(\boldsymbol{U}) + \gamma \frac{K-1}{K} T$$

$$+ \sum_{t=1}^{T} \left( (1-a_t)\mathbb{1}[y_t^\star \neq y_t] + a_t \frac{K-1}{K} - \ell_t(\boldsymbol{W}_t) + \frac{\eta}{2}\|\boldsymbol{g}_t\|^2 \right)$$

$$\leq \frac{\|\boldsymbol{U}\|^2}{2\eta} + \sum_{t=1}^{T} \ell_t(\boldsymbol{U}).$$

Using $\eta = \frac{\ln(2)}{2KX^2}$ completes the proof.

$\square$

**Lemma 2.** *Let $\ell_t$ be the logistic loss* (1)*, then*

$$\|\nabla \ell_t(\boldsymbol{W}_t)\|^2 \leq \frac{2}{\ln(2)}\|\boldsymbol{x}_t\|^2 \ell_t(\boldsymbol{W}_t).$$

*Proof.* We have

$$\|\nabla \ell_t(\boldsymbol{W}_t)\|^2 = \frac{1}{\ln(2)^2}\|\boldsymbol{x}_t\|^2 \left( \sum_{k=1}^{K} (\mathbb{1}[y_t = k] - \tilde{p}_t(k))^2 \right)$$

$$\leq \frac{1}{\ln(2)^2}\|\boldsymbol{x}_t\|^2 \left( \sum_{k=1}^{K} |\mathbb{1}[y_t = k] - \tilde{p}_t(k)| \right)^2$$

$$\leq -2\frac{1}{\ln(2)}\|\boldsymbol{x}_t\|^2 \log_2(\tilde{p}_t(y_t))$$

$$= 2\frac{1}{\ln(2)}\|\boldsymbol{x}_t\|^2 \ell_t(\boldsymbol{W}_t),$$

where the last inquality follows from Pinsker's inequality (Cover and Thomas, 1991, Lemma 12.6.1).

$\square$

## C.2  Details of Section 4.2

*Proof of Theorem 2.* We will prove the Theorem by showing that the surrogate gap is bounded by $0$ and then using Lemma 1. Let $\tilde{k} = \arg\max_{k \neq y_t}\langle \boldsymbol{W}_t^k, \boldsymbol{x}_t \rangle$. The gradient of the smooth multiclass hinge loss is given by

$$\nabla \ell_t(\boldsymbol{W}_t) = \begin{cases} (\boldsymbol{e}_{\tilde{k}} - \boldsymbol{e}_{y_t}) \otimes \boldsymbol{x}_t & \text{if } y_t^\star \neq y_t \\ (\boldsymbol{e}_{\tilde{k}} - \boldsymbol{e}_{y_t}) \otimes \boldsymbol{x}_t & \text{if } y_t^\star = y_t \text{ and } m_t^\star \leq \beta \\ 0 & \text{if } y_t^\star = y_t \text{ and } m_t^\star > \beta. \end{cases}$$

We continue by writing out the surrogate gap:

$$(1-a_t)\mathbb{1}[y_t^\star \neq y_t] + a_t \frac{K-1}{K} - \ell_t(\boldsymbol{W}_t) + \frac{\eta}{2}\|\boldsymbol{g}_t\|^2$$

$$= \begin{cases} m_t^\star + (1-m_t^\star)\frac{K-1}{K} - (1 - m_t(\boldsymbol{W}_t, y_t)) + \eta\|\boldsymbol{x}_t\|^2 & \text{if } y_t^\star \neq y_t \text{ and } m_t^\star \leq \beta \\ (1-m_t^\star)\frac{K-1}{K} - (1-m_t^\star) + \eta\|\boldsymbol{x}_t\|^2 & \text{if } y_t^\star = y_t \text{ and } m_t^\star \leq \beta \\ 1 - (1 - m_t(\boldsymbol{W}_t, y_t)) + \eta\|\boldsymbol{x}_t\|^2 & \text{if } y_t^\star \neq y_t \text{ and } m_t^\star > \beta \\ 0 & \text{if } y_t^\star = y_t \text{ and } m_t^\star > \beta. \end{cases} \tag{10}$$

In the remainder of the proof we will repeatedly use the following useful inequality for whenever $y_t \neq y_t^\star$:

$$m_t^\star + m_t(\boldsymbol{W}_t, y_t) = \langle \boldsymbol{W}_t^{y_t^\star}, \boldsymbol{x}_t \rangle - \max_{k \neq y_t^\star}\langle \boldsymbol{W}_t^k, \boldsymbol{x}_t \rangle + \langle \boldsymbol{W}_t^{y_t}, \boldsymbol{x}_t \rangle - \max_{k \neq y_t}\langle \boldsymbol{W}_t^k, \boldsymbol{x}_t \rangle$$

$$= \langle \boldsymbol{W}_t^{y_t}, \boldsymbol{x}_t \rangle - \max_{k \neq y_t^\star}\langle \boldsymbol{W}_t^k, \boldsymbol{x}_t \rangle \tag{11}$$

$$\leq \langle \boldsymbol{W}_t^{y_t}, \boldsymbol{x}_t \rangle - \langle \boldsymbol{W}_t^{y_t}, \boldsymbol{x}_t \rangle = 0.$$

We now split the analysis into the cases in (10). We start with $y_t^\star \neq y_t$ and $m_t^\star \leq \beta$, in which case the surrogate gap can be bounded by 0 when $\eta \leq \frac{1}{KX^2}$:

$$m_t^\star + (1 - m_t^\star)\frac{K-1}{K} - (1 - m_t(\boldsymbol{W}_t, y_t)) + \eta\|\boldsymbol{x}_t\|^2$$

$$= m_t^\star + m_t(\boldsymbol{W}_t, y_t) + (1 - m_t^\star)\frac{K-1}{K} - 1 + \eta\|\boldsymbol{x}_t\|^2$$

$$\leq -\frac{1}{K} + \eta X^2 \qquad\qquad \text{(by equation (11))}$$

$$\leq 0.$$

We continue with the case where $y_t^\star = y_t$ and $m_t^\star \leq \beta$. In this case we have:

$$(1 - m_t^\star)\frac{K-1}{K} - (1 - m_t^\star) + \eta\|\boldsymbol{x}_t\|^2 = -(1 - m_t^\star)\frac{1}{K} + \eta\|\boldsymbol{x}_t\|^2 \leq -\frac{1-\beta}{K} + \eta X^2,$$

which is zero since $\eta = \frac{1-\beta}{KX^2}$.

Finally, in the case where $y_t^\star \neq y_t$ and $m_t^\star > \beta$ we have:

$$1 - (1 - m_t(\boldsymbol{W}_t, y_t)) + \eta\|\boldsymbol{x}_t\|^2 = m_t(\boldsymbol{W}_t, y_t) + \eta\|\boldsymbol{x}_t\|^2$$

$$\leq -m_t^\star + \eta\|\boldsymbol{x}_t\|^2 \qquad\qquad \text{(by equation (11))}$$

$$\leq -\beta + \eta X^2,$$

which is bounded by zero since $\beta = \frac{1}{K}$ and $\eta \leq \frac{1}{KX^2}$.

We now apply Lemma 1, plug in $\gamma = 0$, and use the above to find:

$$\mathbb{E}\left[\sum_{t=1}^T \mathbb{1}[y_t' \neq y_t]\right] \leq \frac{\|\boldsymbol{U}\|^2}{2\eta} + \sum_{t=1}^T \ell_t(\boldsymbol{U}) + \gamma T$$

$$+ \sum_{t=1}^T \left((1 - a_t)\mathbb{1}[y_t^\star \neq y_t] + a_t\frac{K-1}{K} - \ell_t(\boldsymbol{W}_t) + \frac{\eta}{2}\|\boldsymbol{g}_t\|^2\right)$$

$$\leq \frac{\|\boldsymbol{U}\|^2}{2\eta} + \sum_{t=1}^T \ell_t(\boldsymbol{U}).$$

Using $\eta = \frac{1-\beta}{KX^2} = \frac{K-1}{K^2X^2}$ completes the proof. $\qquad\square$

### C.3 Details of Section 4.3

*Proof of Theorem 3.* We will prove the Theorem by showing that the surrogate gap is bounded by 0 and then using Lemma 1. Let $\tilde{k} = \arg\max_{k \neq y_t}\langle \boldsymbol{W}_t^k, \boldsymbol{x}_t\rangle$. The gradient of the smooth multiclass hinge loss is given by

$$\nabla \ell_t(\boldsymbol{W}_t) = \begin{cases} 2(\boldsymbol{e}_{\tilde{k}} - \boldsymbol{e}_{y_t}) \otimes \boldsymbol{x}_t & \text{if } y_t^\star \neq y_t \\ 2(\boldsymbol{e}_{\tilde{k}} - \boldsymbol{e}_{y_t})(1 - m_t^\star) \otimes \boldsymbol{x}_t & \text{if } y_t^\star = y_t \text{ and } m_t^\star < 1 \\ 0 & \text{if } y_t^\star = y_t \text{ and } m_t^\star \geq 1. \end{cases}$$

We continue by writing out the surrogate gap:

$$(1 - a_t)\mathbb{1}[y_t^\star \neq y_t] + a_t\frac{K-1}{K} - \ell_t(\boldsymbol{W}_t) + \frac{\eta}{2}\|\boldsymbol{g}_t\|^2$$

$$= \begin{cases} 2m_t^\star - m_t^{\star 2} + (1 - m_t^\star)^2\frac{K-1}{K} - (1 - 2m_t(\boldsymbol{W}_t, y_t)) + \eta 4\|\boldsymbol{x}_t\|^2 & \text{if } y_t^\star \neq y_t \text{ and } m_t^\star < 1 \\ (1 - m_t^\star)^2\frac{K-1}{K} - (1 - m_t^\star)^2 + \eta 4\|\boldsymbol{x}_t\|^2(1 - m_t^\star)^2 & \text{if } y_t^\star = y_t \text{ and } m_t^\star < 1 \\ 1 - (1 - 2m_t(\boldsymbol{W}_t, y_t)) + \eta 4\|\boldsymbol{x}_t\|^2 & \text{if } y_t^\star \neq y_t \text{ and } m_t^\star \geq 1 \\ 0 & \text{if } y_t^\star = y_t \text{ and } m_t^\star \geq 1. \end{cases}$$

$$(12)$$

We now split the analysis into the cases in (12). We start with the case where $y_t^\star \neq y_t$ and $m_t^\star < 1$. By using (11) we can see that with $\eta = \frac{1}{4KX^2}$ the surrogate gap is bounded by 0:

$$2m_t^\star - m_t^{\star 2} + (1 - m_t^\star)^2 \frac{K-1}{K} - (1 - 2m_t(\boldsymbol{W}_t, y_t)) + \eta 4\|\boldsymbol{x}_t\|^2$$

$$= 2(m_t^\star + m_t(\boldsymbol{W}_t, y_t)) - m_t^{\star 2} + (1 - m_t^\star)^2 \frac{K-1}{K} - 1 + \eta 4\|\boldsymbol{x}_t\|^2$$

$$\leq -m_t^{\star 2} + (1 - m_t^\star)^2 \frac{K-1}{K} - 1 + \eta 4X^2 \qquad \text{(by equation (11))}$$

$$\leq -\frac{1}{K} + \eta 4X^2 \leq 0.$$

The next case we consider is when $y_t^\star = y_t$ and $m_t^\star < 1$. In this case we have

$$(1 - m_t^\star)^2 \frac{K-1}{K} - (1 - m_t^\star)^2 + \eta 4\|\boldsymbol{x}_t\|^2 (1 - m_t^\star)^2 = -(1 - m_t^\star)^2 \frac{1}{K} + \eta 4\|\boldsymbol{x}_t\|^2 (1 - m_t^\star)^2,$$

which is bounded by 0 since $\eta = \frac{1}{4KX^2}$.

Finally, if $y_t^\star \neq y_t$ and $m_t^\star \geq 1$ then

$$1 - (1 - 2m_t(\boldsymbol{W}_t, y_t)) + \eta 4\|\boldsymbol{x}_t\|^2 = 2m_t(\boldsymbol{W}_t, y_t) + \eta 4\|\boldsymbol{x}_t\|^2$$

$$\leq -2m_t^\star + \eta 4\|\boldsymbol{x}_t\|^2 \qquad \text{(by equation (11))}$$

$$\leq -2 + \eta 4X^2,$$

which is bounded by 0 since $\eta < \frac{1}{2X^2}$. We apply Lemma 1 with $\gamma = 0$ and use the above to find:

$$\mathbb{E}\left[\sum_{t=1}^T \mathbb{1}[y_t' \neq y_t]\right] \leq \frac{\|\boldsymbol{U}\|^2}{2\eta} + \sum_{t=1}^T \ell_t(\boldsymbol{U}) + \gamma \frac{K-1}{K} T$$

$$+ \sum_{t=1}^T \left((1 - a_t)\mathbb{1}[y_t^\star \neq y_t] + a_t \frac{K-1}{K} - \ell_t(\boldsymbol{W}_t) + \frac{\eta}{2}\|\boldsymbol{g}_t\|^2\right)$$

$$\leq \frac{\|\boldsymbol{U}\|^2}{2\eta} + \sum_{t=1}^T \ell_t(\boldsymbol{U}).$$

Using $\eta = \frac{1}{4KX^2}$ completes the proof.

$\square$

## D  Details of Bandit Multiclass Classification (Section 5)

### D.1  Details of Section 5.1

*Proof of Theorem 4.* First, by straightforward calculations we can see that $p_t'(y_t) \geq \frac{(1-\gamma)\exp(-2DX)+\gamma}{K} = \delta$. As in the full information case we will prove the Theorem by showing that the surrogate gap is bounded by 0 and then using Lemma 1. We start by writing out the surrogate gap:

$$\mathbb{E}\left[(1 - a_t)\mathbb{1}[y_t^\star \neq y_t] + a_t \frac{K-1}{K} - \mathbb{E}_t[\ell_t(\boldsymbol{W}_t)] + \frac{\eta}{2}\mathbb{E}_t\left[\|\boldsymbol{g}_t\|^2\right]\right]$$

$$= \mathbb{E}\left[(1 - a_t)\mathbb{1}[y_t^\star \neq y_t] + a_t \frac{K-1}{K} + \log_2(\tilde{p}_t(y_t)) + \frac{\eta}{2\ln(2)^2 p_t'(y_t)}\|(\tilde{\boldsymbol{p}}_t - \boldsymbol{e}_{y_t}) \otimes \boldsymbol{x}_t\|^2\right]$$

$$\leq \mathbb{E}\left[(1 - a_t)\mathbb{1}[y_t^\star \neq y_t] + a_t \frac{K-1}{K} + \log_2(\tilde{p}_t(y_t)) - \frac{\eta}{\ln(2)p_t'(y_t)}X^2 \log_2(\tilde{p}_t(y_t))\right]$$

$$= \begin{cases} \frac{K-1}{K} + \mathbb{E}\left[\log_2(\tilde{p}_t(y_t)) - \frac{\eta}{\ln(2)p_t'(y_t)}X^2 \log_2(\tilde{p}_t(y_t))\right] & \text{if } p_t^\star < 0.5 \\ \mathbb{E}\left[p_t^\star + (1 - p_t^\star)\frac{K-1}{K} + \log_2(\tilde{p}_t(y_t)) - \frac{\eta}{\ln(2)p_t'(y_t)}X^2 \log_2(\tilde{p}_t(y_t))\right] & \text{if } y_t^\star \neq y_t \text{ and } p_t^\star \geq 0.5 \\ \mathbb{E}\left[(1 - p_t^\star)\frac{K-1}{K} + \log_2(p_t^\star) - \frac{\eta}{\ln(2)p_t'(y_t^\star)}X^2 \log_2(p_t^\star)\right] & \text{if } y_t^\star = y_t \text{ and } p_t^\star \geq 0.5, \end{cases}$$

$$(13)$$

where the first inequality is due to Lemma 2.

We now split the analysis into the cases in (13). We start with $p_t^\star < 0.5$. In this case we use $1 \le -\log_2(x)$ for $x \in [0, \frac{1}{2}]$ and obtain

$$\frac{K-1}{K} + \mathbb{E}[\log_2(\tilde{p}_t(y_t)) - \frac{\eta}{\ln(2)p_t'(y_t)}X^2 \log_2(\tilde{p}_t(y_t))]$$

$$\le \mathbb{E}\left[ -\frac{K-1}{K}\log_2(\tilde{p}_t(y_t)) + \log_2(\tilde{p}_t(y_t)) - \frac{\eta}{\ln(2)p_t'(y_t)}X^2 \log_2(\tilde{p}_t(y_t)) \right]$$

$$\le \mathbb{E}\left[ -\frac{K-1}{K}\log_2(\tilde{p}_t(y_t)) + \log_2(\tilde{p}_t(y_t)) - \frac{\eta}{\ln(2)\delta}X^2 \log_2(\tilde{p}_t(y_t)) \right]$$

which is bounded by 0 when $\eta \le \frac{\ln(2)\delta}{KX^2}$.

The second case we consider is when $y_t^\star \ne y_t$ and $p_t^\star \ge 0.5$. In this case we use $x \le -\frac{1}{2}\log_2(1-x)$ for $x \in [0.5, 1]$ and $1 - x \le -\frac{1}{2}\log_2(1-x)$ for $x \in [0.5, 1]$ and obtain

$$\mathbb{E}\left[ p_t^\star + (1 - p_t^\star)\frac{K-1}{K} + \log_2(\tilde{p}_t(y_t)) - \frac{\eta}{\ln(2)p_t'(y_t)}X^2 \log_2(\tilde{p}_t(y_t)) \right]$$

$$\le \mathbb{E}\left[ -\frac{1}{2}\log_2(1 - p_t^\star) - \frac{K-1}{K}\frac{1}{2}\log_2(1 - p_t^\star) + \log_2(\tilde{p}_t(y_t)) - \frac{\eta}{\ln(2)\delta}X^2 \log_2(\tilde{p}_t(y_t)) \right]$$

$$= \mathbb{E}\left[ -\frac{1}{2}\log_2\left( \sum_{k \ne y_t}^{K} \tilde{p}_t(k) \right) - \frac{K-1}{K}\frac{1}{2}\log_2\left( \sum_{k \ne y_t}^{K} \tilde{p}_t(k) \right) + \log_2(\tilde{p}_t(y_t)) - \frac{\eta}{\ln(2)\delta}X^2 \log_2(\tilde{p}_t(y_t)) \right]$$

$$\le \mathbb{E}\left[ -\frac{1}{2}\log_2\left( \tilde{p}_t(y_t) \right) - \frac{K-1}{K}\frac{1}{2}\log_2\left( \tilde{p}_t(y_t) \right) + \log_2(\tilde{p}_t(y_t)) - \frac{\eta}{\ln(2)\delta}X^2 \log_2(\tilde{p}_t(y_t)) \right]$$

$$= \mathbb{E}\left[ \frac{1}{2K}\log_2(\tilde{p}_t(y_t)) - \frac{\eta}{\ln(2)\delta}X^2 \log_2(\tilde{p}_t(y_t)) \right],$$

which is bounded by 0 since $\eta = \frac{\ln(2)\delta}{2KX^2}$.

The last case we need to consider is when $y_t^\star = y_t$ and $p_t^\star \ge 0.5$. In this case we use $1 - x \le -\log_2(x)$ and obtain

$$\mathbb{E}\left[ (1 - p_t^\star)\frac{K-1}{K} + \log_2(p_t^\star) - \frac{\eta}{\ln(2)p_t'(y_t^\star)}X^2 \log_2(p_t^\star) \right]$$

$$\le \mathbb{E}\left[ -\frac{K-1}{K}\log_2(p_t^\star) + \log_2(p_t^\star) - \frac{\eta}{\ln(2)\delta}X^2 \log_2(p_t^\star) \right],$$

which is bounded by 0 when $\eta \le \frac{\ln(2)\delta}{KX^2}$.

We now apply Lemma 1 and use the above to find:

$$\mathbb{E}\left[ \sum_{t=1}^{T} \mathbb{1}[y_t' \ne y_t] \right] \le \frac{\|U\|^2}{2\eta} + \mathbb{E}\left[ \sum_{t=1}^{T} \ell_t(U) \right] + \gamma\frac{K-1}{K}T$$

$$+ \mathbb{E}\left[ \sum_{t=1}^{T} \left( (1 - a_t)\mathbb{1}[y_t^\star \ne y_t] + a_t\frac{K-1}{K} - \ell_t(W_t) + \frac{\eta}{2}\|g_t\|^2 \right) \right]$$

$$\le \frac{\|U\|^2}{2\eta} + \gamma T + \mathbb{E}\left[ \sum_{t=1}^{T} \ell_t(U) \right].$$

Using $\eta = \frac{\ln(2)\delta}{2KX^2}$ gives us:

$$\mathbb{E}\left[ \sum_{t=1}^{T} \mathbb{1}[y_t' \ne y_t] \right] \le \frac{K^2 X^2 \|U\|^2}{\ln(2)((1 - \gamma)\exp(-2DX) + \gamma)} + \gamma T + \mathbb{E}\left[ \sum_{t=1}^{T} \ell_t(U) \right],$$

Setting $\gamma = 0$ gives us

$$\mathbb{E}\left[\sum_{t=1}^{T} \mathbb{1}[y_t' \neq y_t]\right] \leq \frac{K^2 X^2 D^2}{\ln(2)\exp(-2DX)} + \mathbb{E}\left[\sum_{t=1}^{T} \ell_t(\boldsymbol{U})\right].$$

If instead we set $\gamma = \min\left\{1, \sqrt{\frac{K^2 X^2 D^2}{\ln(2)T}}\right\}$ we consider two cases. In the case where $1 \leq \sqrt{\frac{K^2 X^2 D^2}{T}}$ we have that $T \leq K^2 X^2 D^2$ and therefore

$$\mathbb{E}\left[\sum_{t=1}^{T} \mathbb{1}[y_t' \neq y_t]\right] \leq 2\frac{K^2 X^2 D^2}{\ln(2)} + \mathbb{E}\left[\sum_{t=1}^{T} \ell_t(\boldsymbol{U})\right].$$

In the case where $1 > \sqrt{\frac{K^2 X^2 D^2}{T}}$ we have that

$$\mathbb{E}\left[\sum_{t=1}^{T} \mathbb{1}[y_t' \neq y_t]\right] \leq 2KXD\sqrt{\frac{T}{\ln(2)}} + \mathbb{E}\left[\sum_{t=1}^{T} \ell_t(\boldsymbol{U})\right],$$

which after combining the above completes the proof. $\qquad\square$

## D.2  Details of Section 5.2

*Proof of Theorem 5.*  First, note that $p_t'(y_t) \geq \frac{\gamma}{K}$. The proof proceeds in a similar way as in the full information setting (Theorem 2), except now we use that $p_t'(y_t) \geq \frac{\gamma}{K}$ to bound $\mathbb{E}_t[\|\boldsymbol{g}_t\|^2]$. We will prove the Theorem by showing that the surrogate gap is bounded by 0 and then using Lemma 1. We start by splitting the surrogate gap in cases:

$$\mathbb{E}\left[(1 - a_t)\mathbb{1}[y_t^\star \neq y_t] + a_t \frac{K-1}{K} - \mathbb{E}_t[\ell_t(\boldsymbol{W}_t)] + \frac{\eta}{2}\mathbb{E}_t[\|\boldsymbol{g}_t\|^2]\right]$$

$$= \begin{cases} \mathbb{E}\left[m_t^\star + (1 - m_t^\star)\frac{K-1}{K} - (1 - m_t(\boldsymbol{W}_t, y_t)) + \frac{\eta}{p_t'(y_t)}\|\boldsymbol{x}_t\|^2\right] & \text{if } y_t^\star \neq y_t \text{ and } m_t^\star \leq \beta \\ \mathbb{E}\left[(1 - m_t^\star)\frac{K-1}{K} - (1 - m_t^\star) + \frac{\eta}{p_t'(y_t)}\|\boldsymbol{x}_t\|^2\right] & \text{if } y_t^\star = y_t \text{ and } m_t^\star \leq \beta \\ \mathbb{E}\left[1 - (1 - m_t(\boldsymbol{W}_t, y_t)) + \frac{\eta}{p_t'(y_t)}\|\boldsymbol{x}_t\|^2\right] & \text{if } y_t^\star \neq y_t \text{ and } m_t^\star > \beta \\ 0 & \text{if } y_t^\star = y_t \text{ and } m_t^\star > \beta. \end{cases}$$
$$\tag{14}$$

We now split the analysis into the cases in (14). We start with $y_t^\star \neq y_t$ and $m_t^\star \leq \beta$. The surrogate gap can now be bounded by 0 when $\eta \leq \frac{\gamma}{K^2 X^2}$:

$$\mathbb{E}\left[m_t^\star + (1 - m_t^\star)\frac{K-1}{K} - (1 - m_t(\boldsymbol{W}_t, y_t)) + \frac{\eta}{p_t'(y_t)}\|\boldsymbol{x}_t\|^2\right]$$

$$= \mathbb{E}\left[m_t^\star + m_t(\boldsymbol{W}_t, y_t) + (1 - m_t^\star)\frac{K-1}{K} - 1 + \frac{\eta}{p_t'(y_t)}\|\boldsymbol{x}_t\|^2\right]$$

$$\leq -\frac{1}{K} + \frac{K\eta}{\gamma}X^2 \qquad\qquad\qquad\text{(equation (11))}$$

$$\leq 0.$$

We continue with the case where $y_t^\star = y_t$ and $m_t^\star \leq \beta$. In this case we have:

$$\mathbb{E}\left[(1 - m_t^\star)\frac{K-1}{K} - (1 - m_t^\star) + \eta\|\boldsymbol{x}_t\|^2\right] = \mathbb{E}\left[-(1 - m_t^\star)\frac{1}{K} + \frac{\eta}{p_t'(y_t)}\|\boldsymbol{x}_t\|^2\right]$$

$$\leq -\frac{1 - \beta}{K} + \frac{K\eta}{\gamma}X^2,$$

which is bounded by zero since $\eta = \frac{\gamma(1-\beta)}{K^2 X^2}$.

Finally, in the case where $y_t^\star \neq y_t$ and $m_t^\star > \beta$ we have:

$$\mathbb{E}\left[1 - (1 - m_t(\boldsymbol{W}_t, y_t)) + \frac{\eta}{p_t'(y_t)}\|\boldsymbol{x}_t\|^2\right] = \mathbb{E}\left[m_t(\boldsymbol{W}_t, y_t) + \frac{\eta}{p_t'(y_t)}\|\boldsymbol{x}_t\|^2\right]$$

$$\leq \mathbb{E}\left[-m_t^\star + \frac{\eta}{p_t'(y_t)}\|\boldsymbol{x}_t\|^2\right] \quad \text{(by equation (11))}$$

$$\leq -\beta + \frac{K\eta}{\gamma}X^2,$$

which is bounded by zero since $\eta = \frac{\gamma(1-\beta)}{K^2 X^2}$ and $\beta \leq 0.5$.

We now apply Lemma 1 and use the above to find:

$$\mathbb{E}\left[\sum_{t=1}^T \mathbb{1}[y_t' \neq y_t]\right] \leq \frac{\|\boldsymbol{U}\|^2}{2\eta} + \mathbb{E}\left[\sum_{t=1}^T \ell_t(\boldsymbol{U})\right] + \gamma\frac{K-1}{K}T$$

$$+ \mathbb{E}\left[\sum_{t=1}^T \left((1-a_t)\mathbb{1}[y_t^\star \neq y_t] + a_t\frac{K-1}{K} - \ell_t(\boldsymbol{W}_t) + \frac{\eta}{2}\|\boldsymbol{g}_t\|^2\right)\right]$$

$$\leq \frac{D^2}{2\eta} + \gamma\frac{K-1}{K}T + \mathbb{E}\left[\sum_{t=1}^T \ell_t(\boldsymbol{U})\right].$$

Plugging in $\eta = \frac{\gamma(1-\beta)}{K^2 X^2}$ and $\beta = \frac{1}{K}$ gives us:

$$\mathbb{E}\left[\sum_{t=1}^T \mathbb{1}[y_t' \neq y_t]\right] \leq \frac{K^3 X^2 D^2}{2\gamma(K-1)} + \gamma\frac{K-1}{K}T + \mathbb{E}\left[\sum_{t=1}^T \ell_t(\boldsymbol{U})\right].$$

We now set $\gamma = \min\left\{1, \sqrt{\frac{K^3 X^2 D^2}{2(1-\beta)(K-1)T}}\right\}$. In the case where $1 \leq \sqrt{\frac{K^3 X^2 D^2}{2(1-\beta)(K-1)T}}$ we have

$$\mathbb{E}\left[\sum_{t=1}^T \mathbb{1}[y_t' \neq y_t]\right] \leq \frac{K^3 X^2 D^2}{K-1} + \mathbb{E}\left[\sum_{t=1}^T \ell_t(\boldsymbol{U})\right].$$

In the case where $1 > \sqrt{\frac{K^3 X^2 D^2}{2(1-\beta)(K-1)T}}$ we have

$$\mathbb{E}\left[\sum_{t=1}^T \mathbb{1}[y_t' \neq y_t]\right] \leq 2KXD\sqrt{\frac{T}{2}} + \mathbb{E}\left[\sum_{t=1}^T \ell_t(\boldsymbol{U})\right],$$

which completes the proof. $\qquad\square$

### D.3 Details of Section 5.3

*Proof of Theorem 6.* First, note that $p_t'(y_t) \geq \frac{\gamma}{K}$. The proof proceeds in a similar way as in the full information case. We will prove the Theorem by showing that the surrogate gap is bounded by 0 and then using Lemma 1. We start by writing out the surrogate gap:

$$\mathbb{E}\left[(1-a_t)\mathbb{1}[y_t^\star \neq y_t] + a_t\frac{K-1}{K} - \mathbb{E}_t[\ell_t(\boldsymbol{W}_t)] + \frac{\eta}{2}\mathbb{E}_t[\|\boldsymbol{g}_t\|^2]\right]$$

$$= \begin{cases} \mathbb{E}\left[2m_t^\star - m_t^{\star 2} + (1 - m_t^\star)^2\frac{K-1}{K} - (1 - 2m_t(\boldsymbol{W}_t, y_t)) + \frac{\eta}{p_t'(y_t)}4\|\boldsymbol{x}_t\|^2\right] & \text{if } y_t^\star \neq y_t \text{ and } m_t^\star < 1 \\ \mathbb{E}\left[(1 - m_t^\star)^2\frac{K-1}{K} - (1 - m_t^\star)^2 + \frac{\eta}{p_t'(y_t)}4\|\boldsymbol{x}_t\|^2(1 - m_t^\star)^2\right] & \text{if } y_t^\star = y_t \text{ and } m_t^\star < 1 \\ \mathbb{E}\left[1 - (1 - 2m_t(\boldsymbol{W}_t, y_t)) + \frac{\eta}{p_t'(y_t)}4\|\boldsymbol{x}_t\|^2\right] & \text{if } y_t^\star \neq y_t \text{ and } m_t^\star \geq 1 \\ 0 & \text{if } y_t^\star = y_t \text{ and } m_t^\star \geq 1. \end{cases}$$
$$(15)$$

We now split the analysis into the cases in (15). We start with the case where $y_t^\star \neq y_t$ and $m_t^\star < 1$. By using (11) we can see that for $\eta = \frac{\gamma}{4K^2X^2}$

$$\mathbb{E}\left[2m_t^\star - m_t^{\star 2} + (1 - m_t^\star)^2\frac{K-1}{K} - (1 - 2m_t(\boldsymbol{W}_t, y_t)) + \frac{\eta}{p_t'(y_t)}4\|\boldsymbol{x}_t\|^2\right]$$

$$= \mathbb{E}\left[2(m_t^\star + m_t(\boldsymbol{W}_t, y_t)) - m_t^{\star 2} + (1 - m_t^\star)^2\frac{K-1}{K} - 1 + \frac{\eta}{p_t'(y_t)}4\|\boldsymbol{x}_t\|^2\right]$$

$$\leq \mathbb{E}\left[-m_t^{\star 2} + (1 - m_t^\star)^2\frac{K-1}{K} - 1 + \frac{\eta}{p_t'(y_t)}4X^2\right] \qquad \text{(by equation (11))}$$

$$\leq -\frac{1}{K} + \frac{K\eta}{\gamma}4X^2 \leq 0.$$

The next case we consider is when $y_t^\star = y_t$ and $m_t^\star < 1$. In this case we have

$$\mathbb{E}\left[(1 - m_t^\star)^2\frac{K-1}{K} - (1 - m_t^\star)^2 + \frac{\eta}{p_t'(y_t)}4\|\boldsymbol{x}_t\|^2(1 - m_t^\star)^2\right]$$

$$= \mathbb{E}\left[-(1 - m_t^\star)^2\frac{1}{K} + \frac{\eta}{p_t'(y_t)}4\|\boldsymbol{x}_t\|^2(1 - m_t^\star)^2\right]$$

$$= \mathbb{E}\left[-(1 - m_t^\star)^2\frac{1}{K} + \frac{K\eta}{\gamma}4X^2(1 - m_t^\star)^2\right],$$

which is bounded by 0 since $\eta = \frac{\gamma}{4K^2X^2}$.

Finally, if $y_t^\star \neq y_t$ and $m_t^\star \geq 1$ then

$$\mathbb{E}\left[1 - (1 - 2m_t(\boldsymbol{W}_t, y_t)) + \frac{\eta}{p_t'(y_t)}4\|\boldsymbol{x}_t\|^2\right] = \mathbb{E}\left[2m_t(\boldsymbol{W}_t, y_t) + \frac{\eta}{p_t'(y_t)}4\|\boldsymbol{x}_t\|^2\right]$$

$$\leq \mathbb{E}\left[-2m_t^\star + \frac{\eta}{p_t'(y_t)}4\|\boldsymbol{x}_t\|^2\right] \quad \text{(by equation (11))}$$

$$\leq -2 + \frac{K\eta}{\gamma}4X^2,$$

which is bounded by 0 since $\eta < \frac{\gamma}{2K^2X^2}$. We apply Lemma 1 and use the above to find:

$$\mathbb{E}\left[\sum_{t=1}^T \mathbb{1}[y_t' \neq y_t]\right] \leq \frac{\|\boldsymbol{U}\|^2}{2\eta} + \mathbb{E}\left[\sum_{t=1}^T \ell_t(\boldsymbol{U})\right] + \gamma T$$

$$+ \mathbb{E}\left[\sum_{t=1}^T \left((1 - a_t)\mathbb{1}[y_t^\star \neq y_t] + a_t\frac{K-1}{K} - \ell_t(\boldsymbol{W}_t) + \frac{\eta}{2}\|\boldsymbol{g}_t\|^2\right)\right]$$

$$\leq \frac{D^2}{2\eta} + \gamma T + \mathbb{E}\left[\sum_{t=1}^T \ell_t(\boldsymbol{U})\right].$$

Plugging in $\eta = \frac{\gamma}{4K^2X^2}$ gives us:

$$\mathbb{E}\left[\sum_{t=1}^T \mathbb{1}[y_t' \neq y_t]\right] \leq \frac{2K^2X^2D^2}{\gamma} + \gamma T + \mathbb{E}\left[\sum_{t=1}^T \ell_t(\boldsymbol{U})\right].$$

Now we set $\gamma = \min\left\{1, \sqrt{\frac{2K^2X^2D^2}{T}}\right\}$. In the case where $1 \leq \sqrt{\frac{2K^2X^2D^2}{T}}$ we have

$$\mathbb{E}\left[\sum_{t=1}^T \mathbb{1}[y_t' \neq y_t]\right] \leq 4K^2X^2D^2 + \mathbb{E}\left[\sum_{t=1}^T \ell_t(\boldsymbol{U})\right].$$

In the case where $1 > \sqrt{\frac{2K^2X^2D^2}{T}}$ we have

$$\mathbb{E}\left[\sum_{t=1}^T \mathbb{1}[y_t' \neq y_t]\right] \leq 2DKX\sqrt{2T} + \mathbb{E}\left[\sum_{t=1}^T \ell_t(\boldsymbol{U})\right],$$

---

**Algorithm 2** ADAHEDGE with abstention

---
**Input:** ADAHEDGE
  1: **for** $t = 1 \ldots T$ **do**
  2:     Obtain expert predictions $\boldsymbol{y}_t = (y_t^1, \ldots, y_t^d)^\top \in [-1, 1]^d$
  3:     Obtain expert distribution $\hat{\boldsymbol{p}}_t$ from ADAHEDGE
  4:     Set $\hat{y}_t = \langle \hat{\boldsymbol{p}}_t, \boldsymbol{y}_t \rangle$
  5:     Let $y_t^\star = \text{sign}(\hat{y}_t)$
  6:     Set $b_t = 1 - |\hat{y}_t|$
  7:     Predict $y_t' = y_t^\star$ with probability $1 - b_t$ and predict $y_t' = *$ with probability $b_t$
  8:     Obtain $\ell_t$ and send $\ell_t$ to ADAHEDGE
  9: **end for**

---

which completes the proof.

$\square$

## E   Online Classification with Abstention

The online classification with abstention setting was introduced by Neu and Zhivotovskiy (2020) and is a special case of the prediction with expert advice setting Vovk (1990); Littlestone and Warmuth (1994). For brevity we only consider the case where there are only 2 labels, -1 and 1. The online classification with abstention setting is different from the standard classification setting in that the learner has access to a third option, abstaining. Neu and Zhivotovskiy (2020) show that when the cost for abstaining is smaller than $\frac{1}{2}$ in all rounds it is possible to tune Exponential Weights such that it suffers constant regret with respect to the best expert in hindsight. Neu and Zhivotovskiy (2020) only consider the zero-one loss, but we show that a similar bound also holds for the hinge loss (and also for the zero-one loss as a special case of the hinge loss). We use a different proof technique from Neu and Zhivotovskiy (2020), which was the inspiration for the proofs of the mistake bounds of GAPTRON. Instead of vanilla Exponential Weights we use a slight adaptation of ADAHEDGE (De Rooij et al., 2014) to prove constant regret bounds when all abstention costs $c_t$ are smaller than $\frac{1}{2}$. In online classification with abstention, in each round $t$

   1  the learner observes the predictions $y_t^i \in [-1, 1]$ of experts $i = 1, \ldots, d$

   2  based on the experts' predictions the learner predicts $y_t' \in [-1, 1] \cup *$, where $*$ stands for abstaining

   3  the environment reveals $y_t \in \{-1, 1\}$

   4  the learner suffers loss $\ell_t(y_t') = \frac{1}{2}(1 - y_t y_t')$ if $y_t' \in [-1, 1]$ and $c_t$ otherwise.

The algorithm we use can be found in Algorithm 2. A parallel result to Lemma 1 can be found in Lemma 3, which we will use to derive the regret of Algorithm 2.

**Lemma 3.** *For any expert $i$, the expected loss of Algorithm 2 satisfies:*

$$\sum_{t=1}^T \left((1 - b_t)\ell_t(y_t^\star) + b_t c_t\right) \leq \sum_{t=1}^T \ell_t(y_t^i) + \inf_{\eta > 0} \left\{ \frac{\ln(d)}{\eta} + \sum_{t=1}^T \underbrace{\left((1 - b_t)\ell_t(y_t^\star) + c_t b_t + \eta v_t - \ell_t(\hat{y}_t)\right)}_{\text{Abstention gap}} \right\}$$
$$+ \frac{4}{3}\ln(d) + 2,$$

*where $v_t = \mathbb{E}_{i \sim \hat{\boldsymbol{p}}_t}[(\ell_t(\hat{y}_t) - \ell_t(y_t^i))^2]$.*

Before we prove Lemma 3 let us compare Algorithm 2 with GAPTRON. The updates of weight matrix $\boldsymbol{W}_t$ in GAPTRON are performed with OGD. In Algorithm 2 the updates or $\hat{p}_t$ are performed using ADAHEDGE. The roles of $a_t$ in GAPTRON and $b_t$ in Algorithm 2 are similar. The role of $a_t$ is to ensure that the surrogate gap is bounded by 0, the role of $b_t$ is to ensure that the abstention gap is bounded by 0.

*Proof of Lemma 3.* First, ADAHEDGE guarantees that

$$\sum_{t=1}^{T} \ell_t(\hat{y}_t) - \ell_t(y_t^i) \leq 2\sqrt{\ln(d) \sum_{t=1}^{T} v_t + 4/3 \ln(d) + 2}.$$

Using the regret bound of ADAHEDGE we can upper bound the expectation of the loss of the learner as

$$\sum_{t=1}^{T} ((1 - b_t)\ell_t(y_t^\star) + b_t c_t)$$

$$= \sum_{t=1}^{T} \left((1 - b_t)\ell_t(y_t^\star) + b_t c_t + \ell_t(y_t^i) - \ell_t(\hat{y}_t)\right) + \sum_{t=1}^{T} \left(\ell_t(\hat{y}_t) - \ell_t(y_t^i)\right)$$

$$\leq \sum_{t=1}^{T} \left((1 - b_t)\ell_t(y_t^\star) + b_t c_t + \ell_t(y_t^i) - \ell_t(\hat{y}_t)\right) + 2\sqrt{\ln(d) \sum_{t=1}^{T} v_t + 4/3 \ln(d) + 2}$$

$$= \sum_{t=1}^{T} \ell_t(y_t^i) + \inf_{\eta > 0} \left\{ \frac{\ln(d)}{\eta} + \sum_{t=1}^{T} ((1 - b_t)\ell_t(y_t^\star) + c_t b_t + \eta v_t - \ell_t(\hat{y}_t)) \right\} + 4/3 \ln(d) + 2.$$

$\square$

To upper bound the abstention gap by 0 is more difficult than to upper bound the surrogate gap as the negative term is no longer an upper bound on the zero-one loss. Hence, the abstention cost has to be strictly better than randomly guessing as otherwise there is no $\eta$ or $b_t$ such that the abstention gap is smaller than 0. The result for abstention can be found in Theorem 7 below.

**Theorem 7.** *Suppose* $\max_t c_t < \frac{1}{2}$ *for all T. Then Algorithm 2 guarantees*

$$\sum_{t=1}^{T} ((1 - b_t)\ell_t(y_t^\star) + b_t c_t) \leq \sum_{t=1}^{T} \ell_t(y_t^i) + \min \left\{ \frac{\ln(d)}{1 - 2\max_t c_t}, 2\sqrt{\ln(d) \sum_{t=1}^{T} v_t} \right\} + 4/3 \ln(d) + 2.$$

*Proof.* We start by upper bounding the $v_t$ term. We have

$$v_t = \frac{1}{4} \mathbb{E}_{\hat{p}_t} \left[ (y_t^i - \hat{y}_t)^2 \right] \leq \frac{1}{4} (1 - \hat{y}_t)(\hat{y}_t + 1) \leq \frac{1}{2}(1 - |\hat{y}_t|),$$

where the first inequality is the Bhatia-Davis inequality (Bhatia and Davis, 2000). As with the proofs of GAPTRON we split the abstention gap in cases:

$$(1 - b_t)\ell_t(y_t^\star) + c_t b_t + \eta v_t - \ell_t(\hat{y}_t)$$
$$\leq (1 - b_t)\ell_t(y_t^\star) + c_t b_t + \eta \frac{1}{2}(1 - |\hat{y}_t|) - \ell_t(\hat{y}_t) \qquad (16)$$
$$= \begin{cases} c_t(1 - |\hat{y}_t|) + \eta \frac{1}{2}(1 - |\hat{y}_t|) - \frac{1}{2}(1 - |\hat{y}_t|) & \text{if } y_t^\star = y_t \\ |\hat{y}_t| + c_t(1 - |\hat{y}_t|) + \eta \frac{1}{2}(1 - |\hat{y}_t|) - \frac{1}{2}(1 + |\hat{y}_t|) & \text{if } y_t^\star \neq y_t. \end{cases}$$

Note that regardless of the true label $(1 - b_t)\ell_t(y_t^\star) + c_t b_t - \ell_t(\hat{y}_t) \leq 0$ since $c_t < \frac{1}{2}$. Hence, by using Lemma 3, we can see that as long as $c_t < \frac{1}{2}$

$$\sum_{t=1}^{T} (1 - b_t)\ell_t(y_t^\star) + b_t c_t \leq \sum_{t=1}^{T} \ell_t(y_t^i) + 2\sqrt{\ln(d) \sum_{t=1}^{T} v_t + 4/3 \ln(d) + 2}.$$

Now consider the case where $y_t^\star = y_t$. In this case, as long as $\eta \leq 1 - 2c_t$ the abstention gap is bounded by 0. If $y_t^\star \neq y_t$ then

$$|\hat{y}_t| + c_t(1 - |\hat{y}_t|) + \eta \frac{1}{2}(1 - |\hat{y}_t|) - \frac{1}{2}(1 + |\hat{y}_t|) = c_t(1 - |\hat{y}_t|) + \eta \frac{1}{2}(1 - |\hat{y}_t|) - \frac{1}{2}(1 - |\hat{y}_t|).$$

So as long as $\eta \leq 1 - 2c_t$ the abstention gap is bounded by 0. Applying Lemma 3 now gives us

$$\sum_{t=1}^{T}(1 - b_t)\ell_t(y_t^\star) + b_t c_t - \ell_t(y_t^i) \leq \inf_{\eta>0} \left\{ \frac{\ln(d)}{\eta} + \sum_{t=1}^{T}((1 - b_t)\ell_t(y_t^\star) + c_t b_t + \eta v_t - \ell_t(\hat{y}_t)) \right\}$$
$$+ 4/3 \ln(d) + 2$$
$$\leq \frac{\ln(d)}{1 - 2\max_t c_t} + 4/3 \ln(d) + 2,$$

which completes the proof. $\qquad\square$

With a slight modification of the proof of Theorem 7 one can also show a similar result as Theorem 8 by Neu and Zhivotovskiy (2020), albeit with slightly worse constants. We leave this as an exercise for the reader.