[Reviews · NeurIPS 2020]

Review 1

Summary and Contributions: The contribution of this paper is an algorithm for the problem of online multiclass clasification where the agent is presented with a feature vector and must decide the class, both in the full information and bandit case. The main idea behind this algorithm is to update a linear classifier and probabilistically mix the greedy action with a random action with a factor a_t which is a function of the current linear predictor and feature vector. The main idea is to choose the above factor appropriately such that the surrogate gap between the surrogate (convex) loss used as a comparator and the number of mistakes is upper bounded by zero. The contribution is algorithms for different surrogate losses (logistic loss, multiclass hinge loss and smooth multiclass hinge loss) and their regret guarantees. The proposed algortihms improve upon the ones in the literature both in terms of regret and computation time in bandit settings.

Strengths: The idea behind the algorithm (to have a probabilistic selection mixing the greedy estimation with an exploration with a factor that is chosen to minimize the surrogate gap) is novel and interesting. The regret bounds for the bandit feedback setting are better than the state of the art, and the Theorem 4 (independence of the dimension of the feature vector, linear computation time and much better constants for bandit logistic surrogate loss) is an important result for this line of work.

Weaknesses: The guarantees for the full information setting may be worse than the algorithms already known, and it is unknown how probable is that this would be the case (they depend on the norm of the optimal linear predictor, which is hard to analyze). Selecting the mixing factor a_t must be tailored to the surrogate loss function.

Correctness: The proofs for the results seem correct. The paper presents a theoretical result, therefore not empirical methodology is present.

Clarity: The paper is generally well-written, and it is easy to follow the proofs and main ideas.

Relation to Prior Work: The state of the art regarding results on regret and runtime of algortihms for the surrogate losses examined here is clearly discussed and referenced. Table 1 is a nice quick comparison of the results and more detailed discussions are given at the respective sections. It would be nice to discuss a bit more the relation of Neu and Zhivotovskiy and how their ideas are related to the ones developed in the current paper.

Reproducibility: Yes

Additional Feedback: ------ Update ------ Thanks for the additional clarifications on the connections to the literature and the novelty with respect to standard algorithms.


Review 2

Summary and Contributions: This paper considers online multiclass classification, in both the full-information and bandit settings, where the objective is to minimize regret with respect to the hinge loss (or logistic loss or smooth hinge loss) of the best linear predictor in hindsight. The loss of the online algorithm is 0/1-loss, so “bandit feedback” (only learning the loss of the action chosen) corresponds to only learning if the prediction made was correct or not. The paper presents a new algorithm that has a particularly good combination of running time and regret bound in the bandit setting. One thing I was looking for but couldn’t find (maybe I missed it) is a discussion of what makes multiclass special. The gap between loss functions, e.g., as given in Figure 1, holds for binary too. If you set K=2, do your bounds and/or algorithm become the same as a standard approach? Is the challenge for larger K mostly in the bandit setting or even with full information? In fact, it seems that compared to Perceptron you are doing better in the *small* K regime per the discussion on lines 175-180, so why move to multiclass at all? I guess what I’m wondering is: what is the simplest setting (can you give a specific, easy to visualize, distribution?) where GAPTRON beats Perceptron and the other algorithms? In general, there are so many bounds and parameters going around that it’s a little hard to tell what to pay attention to.

Strengths: The paper presents a new algorithm that has a particularly good combination of running time and regret bound for online multi-class classification in the bandit setting.

Weaknesses: It's a crowded space with a lot of different parameters, so it's hard for a reader (or at least for me) to tell exactly what is better when.

Correctness: Appears to be.

Clarity: The paper is written for an expert in online optimization. I wish the paper had more discussion of the regimes one should think about, examples of distributions or adversaries that illustrate key points, etc., to make it easier for a broader audience.

Relation to Prior Work: Yes, though see above.

Reproducibility: Yes

Additional Feedback: Note added: Thanks to the authors for their helpful response. I recommend the authors strive to make the paper more broadly accessible.


Review 3

Summary and Contributions: The paper presents an algorithm called Gaptron for on-line multi-class classification. The bounds of the following kind are provided. The number of classification mistakes is bounded by the surrogate loss of the best regression coefficients plus the regret term. The surrogate loss function is a parameter of the algorithm. Analysis has been performed for full information (where the complete outcome is revealed and sophisticated loss can be calculated) and bandit settings (where only yes/no answers are available, hence 0/1 loss). The algorithm is efficient modulo operations with the surrogate loss (namely, taking the gradient).

Strengths: The paper provides an efficient algorithm with simple form upper bounds. The paper is clear and well-written.

Weaknesses: I think the results of this kind are really interesting when compared against lower bounds. The paper mentions lower bounds once (those of Hazan et al, 2014) and the comparison is not clear. Can more be said on this issue? In particular, I am surprised the dimension of xs does not feature in the bound. Of course it is implicit in X in a way...

Correctness: Yes

Clarity: Yes

Relation to Prior Work: Yes

Reproducibility: Yes

Additional Feedback: Thank you for the answer to my comments. I am keeping my score.


Review 4

Summary and Contributions: This paper present a randomized online learning algorithm (GAPTRON) for multiclass classification, applicable to both the full-information setting and the bandit setting. Three loss functions are analyzed. In the full-information setting, GAPTRON has constant expected regret; in the bandit setting, GAPTRON has the best expected regret with the linear time complexity. The analysis is based on the proposed novel technique exploiting the gap between the 0/1 loss and the surrogate loss.

Strengths: This paper is technically sound. It is a theoretical work without any empirical evaluation.The theoretical result is significant and the proposed technique is novel as far as I know.

Weaknesses: See the additional feedback section for some suggestions.

Correctness: The claims and method make sense to me.

Clarity: Yes, the paper is well written and easy to read.

Relation to Prior Work: Related work is clearly discussed in general. However, since “to be a randomized algorithm is a drawback”, it is not clear whether the discussed related work is deterministic or randomized.

Reproducibility: Yes

Additional Feedback: ## After reading the feedback The feedback is satisfactory to me, and it will be good to clarify the compared deterministic algorithms in the final version. *** I am far from the online learning community, and hence I could only give some suggestions. * The occurrence of y_t in (4), (5) and (6) (after the importance weighting term) is somewhat confusing as y_t is not available to the learner in the bandit setting. I know the notation is correct, but it would be better to replace y_t with y^\prime_t equivalently. * I would like to suggest to add discussion about randomized algorithms and deterministic algorithms when comparing with previous work. It is not clear whether the comparison is fair.

[Author Response · NeurIPS 2020]

Dear referees and chairs,

We would like to thank all referees for their close reading of the manuscript.

Reviewer # 1: We tried to further explain the connection to Neu and Zhivotovskiy in Appendix E, who consider the
Online Classification with abstention setting. Specifically in lines 488 to 492 we try to explain the connection between
our ideas and the results in the Online Classification with abstention setting. However, we see that it is not clear how
the techniques of Neu and Zhivotovskiy relate to Lemma 3 and we will try to clear this up in the final version of the
paper. To clarify, a simplified explanation of the connection to Neu and Zhivotovskiy is that they derive the particular
learning rate for standard Exponential Weights such that the abstention gap in Lemma 3 is bounded by zero.

Reviewer # 2: There is nothing special in particular about the multiclass setting other than that it becomes more difficult
to just guess the correct answer, which is the reason why the number of classes $K$ shows up in the regret bounds of
Gaptron. The first reason to include the multiclass setting is to provide a complete picture of what is possible with our
new technique. The second reason to include the multiclass setting is the bandit setting. Gaptron does not reduce to
a standard algorithm for any $K$, except when a(W, x) = 0 (see also the discussion in lines 115-120). Note that in the
full information setting the regret of Gaptron does not grow as the number of rounds increases, which is a very useful
property since in most applications the number of rounds is very large. As for the discussion in lines 175-181, in the
worst case, the regret bound of the Perceptron is only better when the total number of rounds $T$ is smaller than $K^2$ as
the regret of the Perceptron is $O(\sqrt{T})$ and the regret of Gaptron is $O(K)$. The simplest setting where Gaptron beats
other algorithms is the bandit setting, where the regret of Gaptron is a factor $\sqrt{d}$ smaller than other algorithms, which
also have a higher runtime than Gaptron.

For an overview of the different bounds we provided Table 1. The parameters can be found in Section 2. Importantly,
Gaptron often is on par with, if not better than slower algorithms such as ONS.

Reviewer # 3: Like Reviewer # 1 states we think that the fact that the regret bound of Gaptron does not explicitly
depends on the dimension of the feature vector is a strength rather than a weakness. Indeed, the regret bounds do
depend on the norm of the feature vector, which means that the regret bound implicitly depends on the dimension of the
feature vector. However, the norm of the feature vector is often the preferred measure of the size of the feature vector,
especially when, for example, the elements of the feature vectors are scaled to [-1, 1]. Note that the regret of several
other algorithms depends on both the norm of the feature vector and on the dimension of the feature vector, for example
the algorithm of Foster and Krishnamurthy (2018).

To clarify, the lower bound of Hazan et al (2014) holds for pure logistic loss regret, which is to say that the learner
also suffers logistic loss. This lower bound does not apply to our setting, where the learner suffers the zero-one loss.
Unfortunately, we are only aware of lower bounds in the full information and separable setting, for example Theorem
18 by Foster et. al. (2018). This lower bound shows that with logistic loss the regret bound of Gaptron is tight up to
logarithmic factors.

We would like to point out that our technique to prove the regret of Gaptron is also novel, not just the algorithm. We
hope that this allows future researchers to exploit a similar technique to provide efficient algorithms for other settings.

Reviewer # 4: To clarify, in the full information setting the algorithms with which we compare are deterministic. In the
bandit setting all algorithms are randomized. One of the possible future directions is to derive high probability regret
bounds to better understand the variance of Gaptron. In the final version, we will clarify that the algorithms with which
we compare are deterministic.

Thank you for the suggestion regarding $y_t$ in the bandit setting.

[Meta-Review · NeurIPS 2020]

The paper presents a new algorithm for a problem that is relevant for the conference. Analysis is solid. The paper is unclear in some places, however. Therefore, we strongly urge you to work to make the presentation simpler (e.g., is it possible to simplify the notation in some way to make the paper more "broadly accessible" as suggested by reviewer #2).